# Demystifying Robot Diffusion Policies: Action Memorization and a Simple Lookup Table Alternative

**Chengyang He**[1,2,*]   **Xu Liu**[1,*]   **Gadiel Sznaier Camps**[1]   **Joseph Bruno**[3]
**Guillaume Sartoretti**[2]   **Mac Schwager**[1]

[1]Stanford University   [2]National University of Singapore
[3]Temple University   [*]Equal Contribution

hecy@stanford.edu,liuxujsw@stanford.edu
gsznaier@stanford.edu,brunoj6@temple.edu
guillaume.sartoretti@nus.edu.sg,schwager@stanford.edu

## Abstract

Diffusion policies for visuomotor robot manipulation tasks achieve remarkable dexterity and robustness while only training on a small number of task demonstrations. However, the reason for this performance remains a mystery. In this paper, we offer a surprising hypothesis: diffusion policies essentially memorize an action lookup table—*and this is beneficial*. We posit that, at runtime, diffusion policies find the closest training image to the test image in a latent space, and recall the associated training action (i.e. action chunk), offering reactivity without the need for action generalization. This is effective in the sparse data regime, where there is not enough data density for the model to learn action generalization. We support this claim with systematic empirical evidence, showing that even when conditioned on highly out of distribution (OOD) images, Diffusion Policy still outputs an action chunk from the training data. We evaluate and compare three representative policy families on the same data set: Diffusion Policy, Action Chunking with Transformers (ACT), and GR00T, a pre-trained generalist Vision-Language-Action (VLA) model. We show that Diffusion Policy gives strong action memorization giving surprising robustness in OOD regimes, ACT shows action interpolation with poor robustness in OOD regimes, and GR00T (benefiting from substantial pre-training) shows both action interpolation and OOD robustness. As a simple alternative to Diffusion Policy, we introduce the Action Lookup Table (ALT) policy, showing that an explicit lookup table policy can perform comparably in this low data regime. Despite its simplicity, ALT attains Diffusion Policy–level performance while also providing faster inference and explicit OOD detection via latent-distance thresholds. These results reframe diffusion policies for robot manipulation as reactive memory retrieval under data sparsity, and provide practical tools for interpreting, evaluating, and monitoring such policies. More information can be found at: https://stanfordmsl.github.io/alt/.

## 1 Introduction

Imitation learning for robot manipulation requires training a policy to map from image inputs to action sequence outputs given a relatively small number of demonstrations. Recently, the Diffusion Policy Chi et al. (2023) has emerged as a powerful approach to this problem by modeling the robot's policy as a denoising diffusion probabilistic model Ho et al. (2020). The Diffusion Policy infers an action chunk (a short action sequence, typically 16 time steps long) conditioned on the robot's camera views and joint angles. To infer an action chunk, the policy iteratively denoises a random action chunk using a learned denoising filter. The denoised action chunk is applied to the robot (typically executing 8 of the 16 actions in the chunk), and the loop repeats. The architecture is derived directly from a denoising diffusion model for image generation Saharia et al. (2022b); Rombach et al. (2022), adapted to produce action chunk outputs. The primary advantage of the Diffusion

Policy lies in its ability to model multi-modal action distributions, scale to high-dimensional robot action spaces, and produce long-horizon action sequences. Indeed, recent studies have shown that diffusion policies outperform many existing methods on challenging manipulation benchmarks Ze et al. (2024); Wang et al. (2024a).

The performance of the Diffusion Policy is unquestionable, however the explanation for this performance remains elusive. In particular, typical diffusion policies are trained on between 50 to 200 task demonstrations (small amounts of data), while maintaining the same number of parameters (typically over 100 million) as image generation models trained on billions of images Saharia et al. (2022b); Wikipedia contributors (2024). Furthermore, the common practice is to train diffusion policies until the training loss is low, but the test loss is high — the classic signal for over-fitting. Typically overfitting is associated with poor test-time performance and poor generalization. Yet, it is observed that this overfitting is actually necessary for strong test time performance of the Diffusion Policy. The natural question arises:

*Why do diffusion policies trained to overfit small data sets appear to give strong test-time performance in robot manipulation?*

In this paper we show that, indeed, diffusion policies highly overfit the training data, such that they essentially recall training action chunks at inference. They exhibit little generalization in the action space, neither interpolation nor extrapolation. They effectively perform a lookup table that maps runtime images to training action chunks. Combined with online closed-loop execution with runtime images, this action chunk memorization appears to be a winning recipe for strong manipulation policies obtained from small amounts of demonstration data. We show that for inference in interpolation regimes, in extrapolation regimes, and in highly out-of-distribution (OOD) regimes the diffusion model always recalls action chunks seen at training, giving the policy a surprising robustness in these OOD regimes. We verified this through pick-and-place experiments on a real robot and Can and Square experiments in Robomimic (more details can be found in Appendix D).

It is then reasonable to wonder whether action memorization a property shared by all imitation learning architectures when trained on small data sets, which motivates our second research question:

*Do other imitation learning architectures for robot manipulation also exhibit action memorization when trained on the same dataset?*

We find that an Action Chunking Transformer Zhao et al. (2023); Wu et al. (2024) (ACT) policy trained on the same dataset exhibits action generalization, often producing action chunks that are a blend of several of the training action chunks. However, in OOD regimes ACT infers wild action chunks quite different from the training data, and therefore lacks the robustness of the Diffusion Policy. We also find that GR00T Bjorck et al. (2025), a generalist pre-trained Vision-Language-Action policy, when fine-tuned on the same data also exhibits action generalization, but maintains robustness in OOD regimes, likely benefiting from the large volume of pre-training data.

A key drawback of the Diffusion Policy is slow inference time, which leads to slow robot execution punctuated by pauses as the model recomputes the next inference at the end of each action chunk. This motivates our third research question:

*Can the same action chunk recall behavior be accomplished with a simpler, faster model architecture to yield faster runtime performance?*

To answer this question, we propose a simple lookup table policy with a trained image encoder to map from images to action chunks, which we call the Action Lookup Table (ALT) policy, illustrated in Fig. 1. The ALT policy has similar task success and robustness properties as the Diffusion Policy, while being 300 times faster at inference, and requiring less than 1/100th the memory footprint. Our ALT policy is simpler than previous parametric interpolation-based imitation learning architectures Pari et al. (2021); Sridhar et al. (2023; 2024b), as it does not attempt any interpo-

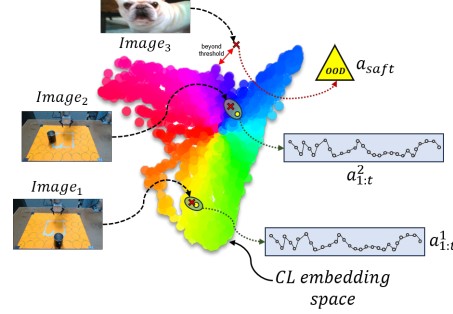

Figure 1: Resulting latent space of our constrastive learning (CL) based ALT on our training data, illustrating the distribution of training examples and example in- (1,2) and out-of-distribution (3) test images.

lation in action space. We obtain multi-modality, similar to the diffusion policy, through a stochastic k-nearest neighbor action chunk lookup.

Concretely, we design a lightweight image-joint pose encoder that maps each provided observation into a low-dimensional feature representation (LFR). LFRs seen during training are labeled with their associated action chunks, which are stored in a lookup table. At deployment time, new observations are encoded to their LFRs, the nearest training LFR is found (with a k-d tree), and the associated action chunk is recalled from the lookup table. The encoder is trained using a contrastive learning objective Chen et al. (2020), encouraging positive sample pairs to be closer while pushing negative pairs apart in the latent space. ALT avoids the costly iterative denoising steps of the Diffusion Policy, enabling faster much inference and much smaller memory footprint.

Our **contributions** are as follows: **(1)** We hypothesize the Diffusion Policy implicitly memorizes an action lookup table. We provide conceptual intuition for this hypothesis and support it through extensive empirical validation. For baselines, we compare to two other imitation learning models, ACT and GR00T-N1.5, and show they provide comparatively more action generalization than Diffuion Policy. **(2)** We propose a simple *Action Lookup Table (ALT)* policy (Fig. 1) that explicitly indexes the nearest training action chunk from an observation embedding. We show that ALT delivers similar performance to a Diffusion Policy, while giving faster inference and requiring less memory. We also propose an explicit OOD monitor for ALT using a distance threshold in the latent space.

## 2 RELATED WORK

Diffusion models, trained by gradually adding Gaussian noise to data during training Ho et al. (2020); Ramesh et al. (2021); Blattmann et al. (2023), were originally developed for high-dimensional data generation tasks such as image, video, or audio synthesis Nichol & Dhariwal (2021); Rombach et al. (2022). These models can produce seemingly novel high-quality images and videos in a variety of different styles through simple text Ramesh et al. (2021); Ruiz et al. (2023); Balaji et al. (2022); Saharia et al. (2022b) and image Saharia et al. (2022a); Tumanyan et al. (2023); Ceylan et al. (2023) prompt conditioning. In order to capture the complex multi-modal distributions inherent in visual and auditory data, these models are often large, containing from hundreds of millions to billions of parameters Saharia et al. (2022b); Wikipedia contributors (2024), and are trained over large datasets with hundreds of millions to billions of examples Rombach et al. (2022).

Leveraging the strong performance of diffusion models, the Diffusion Policy Chi et al. (2023) achieves state-of-the-art performance in visuomotor control for single skill imitation learning. Trained with a limited number of expert demonstrations, the model learns to predict a sequence of robot actions Chi et al. (2023); Ren et al. (2024a); Lu et al. (2024); Lee & Kuo (2024) conditioned on a given observation. This observation can be images Chi et al. (2023), point clouds Ze et al. (2024), semantic labels Wang et al. (2024b); Li et al. (2024) or potential fields Mizuta & Leung (2024). Due to its apparent robustness to perturbations, diffusion policies have been deployed for a wide range of robotics tasks, including manipulation Black et al. (2023); Kim et al. (2022); Chi et al. (2023), multi-skill learning Chen et al. (2023a); Xu et al. (2023), and motion planning Shaoul et al. (2024); Serifi et al. (2024); Sridhar et al. (2024a). Diffusion models have also been used in robotics for data augmentation Kapelyukh et al. (2023); Chen et al. (2023b) to aid in the training of other models.

The phenomenon of memorization in diffusion models has been well-studied in image generation, but not in robotics, to our knowledge. Gu et al. (2023) observed that smaller datasets are prone to cause memorization, especially when conditioned with uninformative labels, while Somepalli et al. (2023a) discovered that reconstructive memorization occurs even for models trained on enormous datasets, with as much as 2% of the generated images being duplicates of the training data. Similarly, Carlini et al. (2023) demonstrated a way to extract known training examples from state-of-the-art models, such as DALL-E 2 Ramesh et al. (2022). Meanwhile, Gu et al. (2023) notes that the traditional denoising score matching objective used during training has a closed-form optimal solution that can only replicate training images. Although, this can be mitigated with synthetic data augmentation Xue et al. (2025), Jain et al. (2024) posits that the denoising process causes diffusion models to learn an attraction basin for each training sample, thereby guiding prompt-conditioned generated images towards memorized data. Wen et al. (2024) corroborates this by noting that diffusion models tend to converge to a known training sample regardless of initialization, suggesting memorization of

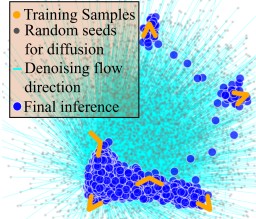 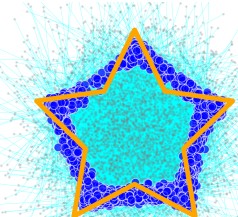 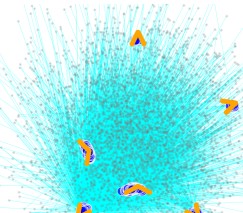 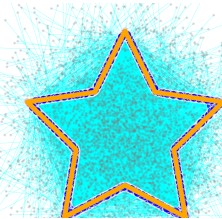

(a) low-capacity model, small data (SD)  (b) low-capacity model, large data (LD)  (c) high-capacity model, SD (e.g. Diffusion Policy)  (d) high-capacity model, LD (e.g. Image Diffusion)

Figure 2: Training a diffusion model from 2D points uniformly distributed on a star-shaped 1D manifold. Each subplot shows a different training regime: (a) A low-capacity model (∼400 parameters) trained on a small dataset (3k samples) gives erratic inferences. (b) The low-capacity model trained on a large dataset (100k samples) generalizes to the wrong manifold. (c) A high-capacity model (∼9.5 million parameters) trained on a small dataset (the Diffusion Policy regime) approximately memorizes the dataset, but does not generalize. All the inferred outputs (blue) overlay the training data (orange) points, essentially implementing a lookup table. (d) A high-capacity model trained on a large dataset shows strong generalization to the correct data manifold (regime of large scale image diffusion models). See C for further discussion.

both the prompt and the denoising trajectory. Similarly, both Somepalli et al. (2023b) and Chen et al. (2024b) indicate that, although less prevalent than conditioned models, memorization still occurs in unconditioned models and Hintersdorf et al. (2024) finds that memorized data are often associated with corresponding individual neurons.

Modifying the loss function Chen et al. (2024a), gradients Chen et al. (2024a), conditioning approach Jain et al. (2024), or keyword prompts Wen et al. (2024); Somepalli et al. (2023b); Ren et al. (2024b) during training and inference are all typical methods for reducing model memorization. However, we propose that while memorization is undesirable for image generation due to privacy and copyright concerns, it is actually beneficial for robotics. When the diffusion model is used in domains with rich input space (e.g., images) but limited output space (e.g., robot actions), the gap between model capacity and output dimensionality, combined with the use of imitation learning (which inherently lacks task-level supervision), makes overfitting via memorization a plausible explanation for its strong performance in in-distribution settings. Along these lines, previous approaches have explored reusing past demonstrations in conjunction with parametric generalizations. They obtain the next action using frame-level nearest-neighbor action retrieval Pari et al. (2021), action prototypes obtained based on a bounded residual Sridhar et al. (2023), or by utilizing a learned context embedding composed of multiple trajectory fragments Sridhar et al. (2024b). In comparison, our proposed ALT method directly obtains the closest action sequence from a lookup table via a contrastive image encoder, bypassing any parametric policy entirely, thereby allowing it to be lightweight and fast unlike more compute-intensive hybrid alternatives.

## 3 DIFFUSION POLICY ANALYSIS

### 3.1 PRELIMINARIES

The output, $\mathbf{x}^0$, of a diffusion model, $\varepsilon_\theta$, is obtained by iteratively removing noise (i.e. denoising) from a starting value, $\mathbf{x}^k$, sampled from a Normal Distribution, $\mathcal{N}(0, \sigma^2 I)$. The denoising process evolves according to

$$\mathbf{x}^{k-1} = \alpha(\mathbf{x}^k - \gamma\varepsilon_\theta(\mathbf{x}^k, k) + \mathcal{N}(0, \sigma^2 I) \tag{1}$$

to remove the noise in $k$ steps based on a predetermined noise schedule that specifies the values of $\alpha, \sigma$, and $\gamma$ at each iteration. This procedure can be thought of as a single stochastic gradient descent step $x' = x - \gamma\nabla E(x)$, where the model $\varepsilon_\theta$ is used to predict the gradient field $\nabla E(x)$. A more detailed explanation of the denoising process can be found in Chi et al. (2023) and Ho et al. (2020).

### 3.2 DIFFUSION MODEL GENERALIZATION REGIMES

We illustrate four generalization regimes for a simple Multi-Layer Perceptron (MLP)-based diffusion denoising model trained to learn a ground-truth distribution consisting of 2D points uniformly sampled on a 1D manifold shaped as a star (Fig. 2). We show qualitative model performance with

a low-capacity vs high-capacity MLP, trained with small vs large data sets. As expected, when a low-capacity model is trained on a small dataset (Fig. 2a), it fails to fit the data adequately. Similarly, due to its limited capacity, when such a model is given sufficient data (Fig. 2b), it is only able to learn an approximation that oversimplifies the data manifold (here, approximating a star shape as a hexagon). In comparison, when a high-capacity model is trained on a small dataset (Fig. 2c), the diffusion model tends to memorize the individual training samples rather than generalizing or interpolating between them. This memorization allows accurate fitting of the limited training points (good for robot manipulation tasks), but results in the model failing to capture the broader underlying data manifold, a behavior that is consistent with our findings for diffusion policies for robot manipulation. This phenomenon is related to manifold overfitting Loaiza-Ganem et al. (2022): when a powerful generative model is trained on data lying on a narrow sub-manifold, it might fit the data too closely while struggling outside that sub-manifold. When the model is provided sufficient data (Fig. 2d), it is now able to effectively fit both the data and the true underlying distribution, representing the regime common in large scale image generation models. However, acquiring large-scale expert demonstrations for robot manipulation that evenly and densely cover the action sequence space remains a significant practical challenge. As Diffusion Policies are trained on larger and larger datasets, they may move toward the large data regime (Fig. 2d) with true generalization on the action manifold, but this seems to be beyond the current state of the art.

## 3.3 Hypothesis and Experiments

The core hypothesis of this paper is that *the impressive reactivity, multimodality, and robustness exhibited by the Diffusion Policy stems not from a deep understanding of the physical task, but from the simple ability to memorize training action chunks and recall an appropriate action chunk when prompted with an image*. To evaluate this hypothesis, we designed a series of cup grasping experiments.[1] We trained a diffusion policy for cup grasping using the standard codebase from Chi et al. (2023), trained with 30/120 demonstrations[2] of cup locations evenly spaced throughout the workspace, with a held-out square in the middle, as indicated by the green circles and blue tape in Fig. 3. The robot has a third-person view fixed camera and a wrist mounted camera, both used to condition the policy. For each position, we performed one demonstration (to remove the confounding effect of multi-modal action generation). We then validated the learned policy on the 30 in-distribution cases, confirming its ability to reproduce the training demonstrations.

To further investigate the action generalization behavior of the policy, we systematically introduced a variety of interpolation and extrapolation inputs, ranging from in-distribution (InD) to out-of-distribution (OOD) and analyzed the resulting behavior. Specifically, we designed four scenarios: (**1:InD-Interpolate**) Placing the cup at evenly spaced test positions located between the original training positions (Fig. 3 green border); (**2:OOD-Interpolate**) Slowly moving the cup from one in-distribution position through an OOD region (blue tape square) to another in-distribution position (Fig. 3 blue border); (**3:OOD-Extrapolate**) Gradually moving the cup from an in-distribution position to an OOD location outside the fixed camera's field of view (Fig. 3 purple border). (**4:OOD-Distractors**) Introducing OOD visual distractors of varying difficulty levels (Fig. 3 yellow bor-

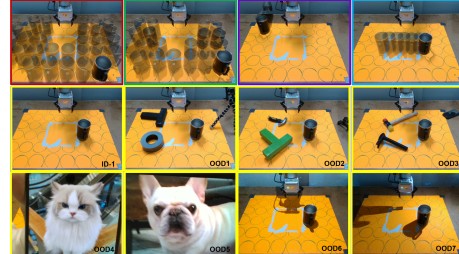

Figure 3: The red outlined panel shows InD tests. The green outlined panel shows InD interpolation tests, with cups evenly placed between training positions. Purple and blue outlined panels illustrate cases where the cup is gradually moved from an in-distribution location to an OOD position. The remaining yellow outlined panels introduce OOD image distractors to assess the model's robustness.

der), including wildly OOD images of a cat and a dog; These settings allowed us to explore and analyze the generalization behavior and potential memory-driven characteristics of the Diffusion Policy. If the Diffusion Policy were performing action generalization, one would expect the following in each scenario: (1:InD-Interpolate) interpolation in the action space; (2:OOD-Interpolate) some action interpolation with degraded performance in the middle, where it is far from the train-

---

[1]We also verified the hypothesis in the simulation environment provided by Robomimic. For details, see Section 3.5 Appendix D.2.

[2]We also conducted training with the standard scale set of 120 demonstrations, but for visualization we present results using a smaller subset.

ing examples; (3:OOD-Extrapolate) progressively degraded action performance as the object moves farther from the training set; and (4:OOD-Distractors) degraded action performance as the number and severity of distractors grows, with dog and cat inducing erratic action sequences.

*In fact, all of these behavioral expectations are incorrect. In every case, the Diffusion Policy almost exactly reproduces one of the training action sequences* as explained below. This is consistent with our action lookup table hypothesis.

### 3.4 REAL-WORLD RESULTS

In this subsection, we introduce a custom metric as a memory-audit, designed to quantify how closely an inference action sequence resembles sequences from the training set. It is defined as: $\mathcal{S} = 1 - \frac{s(\tau^{(r)}, \tau^{(1)})}{s(\tau^{(1)}, \tau^{(2)})}$, where $s(\tau^{(r)}, \tau^{(1)})$ denotes the average Euclidean distance between the matched points on the current action sequence and its closest training sequence, and $s(\tau^{(1)}, \tau^{(2)})$ denotes the distance between the second-closest and the closest training action sequence. If an action sequence closely follows a specific training sequence while maintaining a clear separation from other nearby sequences, this provides strong evidence of memory-based retrieval rather than action generalization. Note that this similarity metric does not measure action *quality*, just action recall. For example, the robot may infer an ineffective action chunk, but if it closely matches one of the training action chunks, the similarity score will be high. We make no claims on the effectiveness of the actions generated by the policies; they are simply recalled from the training data.

#### 3.4.1 DIFFUSION POLICY

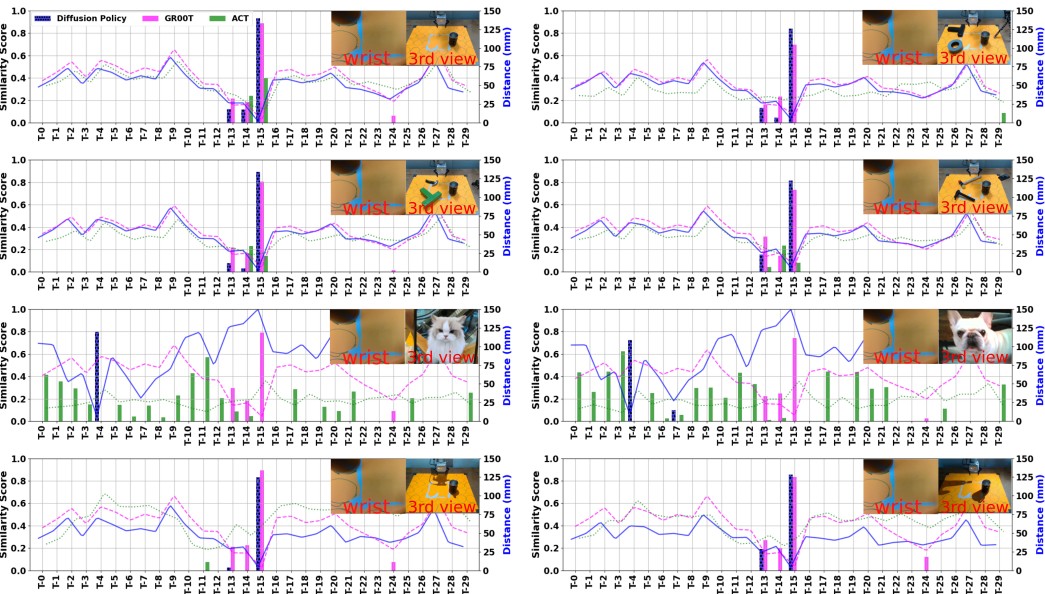

Figure 4: Similarity and distance statistics between each inference trajectory and the training demonstrations for Diffusion Policy, ACT, and GR00T-N1.5. Bars indicate similarity scores, where sharper (more one-hot–like) distributions reflect stronger alignment with a specific training example, signaling stronger memorization behavior. From this comparison, Diffusion Policy exhibits the strongest action memorization, even under completely OOD conditions.

Our experimental results provide compelling evidence supporting the hypothesis that the Diffusion Policy exhibits memory-based action cloning behavior. We first validate this in the in-distribution (InD) setting, where cups are placed exactly as they were during training. In this case, each action sequence almost perfectly overlaps with the corresponding training sequence (see Table 1), where the memory-audit $\mathcal{S}$ close to $1$. This indicates that the model is essentially replaying an action sequence memorized during training when presented with familiar inputs. This behavior persists even under visual OOD scenarios, where we introduced distractors to the environment (OOD-Distractors) as shown in Fig. 3 by the yellow border. Fig. 4 provides a global view of similarity scores across all training trajectories under OOD scenarios. In the presence of distractors, almost all high-similarity

matches are sharply concentrated on a single training trajectory, indicating a surprising OOD default behavior. The diffusion model seems to revert to one or two fallback action sequences when presented with OOD images. Even when the input is entirely unrelated to the task, for example, an image of a cat or a dog, the diffusion model still produces an action sequence that closely resembles one from the training set. We believe these results show that the Diffusion Policy's decision-making is largely governed by memory retrieval, rather than by generalized reasoning over the action space. In both clean and distractor scenarios, the model demonstrates consistent action replay behavior, supporting our hypothesis that its decision-making is fundamentally memory-driven. Additional analyses and results also support this observation. In the InD-Interpolate cases, the Diffusion Policy outputs a tra-

Table 1: Average maximum similarity scores for each method across the various scenarios. Highest possible value is 1 (perfect match with a training action chunk) and the lowest is 0 (far from all training action chunks). In OOD-Extrapolate (side) the cup is moved leftward out of the workspace and in OOD-Extrapolate (back) is when the cup is moved backwards. Note that since ALT directly selects a trajectory from the dataset, it always has a similarity of 1.

| Scenarios | ALT | DP | ACT | GR00T |
|---|---|---|---|---|
| InD | 1 | 0.935 | 0.465 | 0.783 |
| OOD-Distractors | 1 | 0.837 | 0.278 | 0.798 |
| OOD-Interpolate | 1 | 0.690 | 0.406 | 0.725 |
| OOD-Extrapolate (side) | 1 | 0.838 | 0.428 | 0.463 |
| OOD-Extrapolate (back) | 1 | 0.875 | 0.355 | 0.745 |

jectory that closely matches one of the four corresponding nearest-neighbor training trajectories, while in the OOD-Extrapolate cases the generated trajectories are still similar to the training trajectories even once the cup is no longer in view of the 3rd person camera (see Table 1). See Appendix D for more detailed InD visualization results and other OOD results.

### 3.4.2 ACTION CHUNKING WITH TRANSFORMERS

We evaluate *Action Chunking with Transformers (ACT)* Zhao et al. (2023) on the same datasets, test scenarios (including InD and OOD cases). ACT already exhibits a notable degree of interpolation in InD scenarios, which are typically where memorization effects are most evident. In scenarios that are visually in distribution, but require OOD action generalization, the ACT policy seems to sensibly compose action chunks to generalize its behavior. The 3D plot reveals clear evidence of interpolation: several inference trajectories fall cleanly between nearby training trajectories in the correct region.

However, in visual OOD regimes, when there are distractors, ACT produces almost random behavior, which is quite different from Diffusion Policy. As shown in Fig. 4, the model failed to reproduce the correct trajectory (`traj15`). Surprisingly, with distractor inputs (e.g., cats or dogs), the Diffusion Policy returns a trajectory closely resembling some training demonstration (albeit not the correct `traj15`). In comparison, ACT's outputs are essentially random. For additional visualizations, including detailed InD results and further OOD analyses, see Appendix D.

### 3.4.3 GR00T-N1.5

Under full InD conditions, GR00T-N1.5 Bjorck et al. (2025) performs correct inference, consistently matching the correct trajectory. However, GR00T-N1.5 displays mixed behavior in InD-Interpolate: it sometimes interpolates trajectories between trained spots, while at other times it reverts to memorization. In more challenging OOD-Interpolate cases, GR00T-N1.5 does not memorize like the Diffusion Policy; instead, it produces trajectories that lie between training demonstrations. In OOD-Extrapolate, when the cup moves outside the workspace and beyond view, GR00T-N1.5 initially tracks correctly but then collapses to average-like predictions, indicating a lack of systematic extrapolation capability.

In the OOD-Distractors scenario, GR00T-N1.5 demonstrates strong robustness, likely stemming from its extensive vision-language-action pre-training. Unlike the Diffusion Policy, GR00T-N1.5 successfully produces the correct trajectory (`traj15`) under such conditions. It does so by relying on consistent task-relevant cues from the first-person camera while ignoring irrelevant third-person distractor images. Fig. 4 illustrates GR00T-N1.5's behavior under OOD-Distractors: the close alignment between the predicted trajectory and `traj15`, together with the consistent similarity score distribution across distractors, provides strong evidence of robustness. In contrast, both ACT and the Diffusion Policy failed to produce the correct inference trajectory in some of these OOD cases. This indicates that GR00T-N1.5 relies on task-relevant cues (e.g., first-person view of the cup), likely thanks to its large-scale pretrained Vision-Language Model (VLM), rather than overfitting to

spurious correlations, making it more reliable in visually perturbed environments. For other more detailed results, see Appendix D.

### 3.4.4 ANALYSIS

Across the same pick-and-place task, we observe three distinct behaviors from the Diffusion Policy, ACT, and GR00T-N1.5. *Diffusion Policy* acts like a retrieval controller in low-data settings: on InD scenes it effectively replays a specific demo, and in the case of OOD, even when faced with a completely incomprehensible scenario, it will still consider selecting one or two demonstrations from the training data as the output. *ACT* shows less action memorization compared to the Diffusion Policy: its chunked decoder with a smooth latent prior blends action segments, which can appear reasonable in InD-Interpolate and OOD-Interpolate but, under strong distractors, drifts toward prior or average chunks, essentially a hallucinated interpolation. *GR00T-N1.5* exhibits a mix of different behaviors: its diffusion-based action head retains a manifold-attraction effect (drawing outputs toward training-set modes, which can cause memorization), while its VLM (pretrained on internet-scale data) provides task-relevant invariances that resist spurious cues, resulting in better OOD performance. Freezing the high-capacity Diffusion Transformer during fine-tuning on small datasets avoids strong action memorization while keeping the model performant and robust, since the pretrained VLM anchors it with stable conditioning. However, once the Diffusion Transformer itself is fine-tuned, the model shifts toward memorizing specific actions. We speculate that score-based models (e.g., DDPM Ho et al. (2020)) denoise by moving samples toward high-density regions of the conditional data distribution. Mathematically, with $x_t = \alpha_t x_0 + \sigma_t \varepsilon$, Tweedie's formula Efron (2011) gives

$$\hat{x}_0(x_t) = \frac{1}{\alpha_t}(x_t + \sigma_t^2 \nabla_{x_t} \log p_t(x_t|\text{cond})) \approx \frac{x_t - \sigma_t \varepsilon_\theta(x_t, t)}{\alpha_t}, \qquad (2)$$

and the reverse-time SDE Song et al. (2020) is

$$dx = [f(x,t) - g^2(t)\nabla_x \log p_t(x|\text{cond})]dt + g(t)d\bar{\omega}. \qquad (3)$$

Iterating these steps yields a *manifold attraction* effect: samples drift toward modes the diffusion policy's conditional distribution $p(\mathbf{A}_t|\mathbf{O}_t)$. In small-data regimes, those modes can degenerate into mixtures of narrow kernels around demonstrations, so generations tend to stick near particular demos (a memory/replay effect).

Using the same hardware setup as for ACT and Diffusion Policy (RTX 4090), we kept the Diffusion Transformer frozen to avoid out-of-memory issues, following the official GR00T-N1.5 recommendation (`--no-tune_diffusion_model`). In the above experiments, only the small adapter layers and embodiment-specific action heads (a few million parameters) were fine-tuned, while both the VLM and the action expert remained frozen. However, we also fine-tuned the entire action expert (which includes the diffusion transformer), which led to much stronger action memorization compared to runs where the diffusion transformer was kept frozen. We believe that even large diffusion-based architectures tend to memorize when fully tuned or trained from scratch on small datasets. For pretrained VLA models, freezing most of the system and tuning only a small set of parameters can achieve a good balance between generalization/interpolation and in-distribution performance—especially when fine-tuning on small datasets.

### 3.5 SIMULATION RESULTS

In addition to our physical robot experiments, we further validated our hypothesis on the Robomimic benchmark using the official training demonstrations, evaluation trajectories, and pretrained checkpoints provided in the original Diffusion Policy release. As summarized in Table 2, we observe that, except for the Lift task, the rollout trajectories generated by Diffusion Policy exhibit a high degree of similarity to their training demonstrations. This consistent overlap strongly supports our claim that the model's impressive performance often arises from memorization rather than true generalization across unseen states. The comparatively lower similarity observed in the Lift task can be attributed to its simpler nature and the fact that it was trained for substantially fewer epochs in the official setup (only 300 and 450 epochs).

Overall, our findings reveal a systematic tendency of Diffusion Policies to reproduce familiar action patterns encountered during training, rather than synthesizing novel behaviors. Importantly, this

| | Epochs | # Demos | Image Obs. | Epochs | # Demos | Low Dimensional |
|---|---|---|---|---|---|---|
| Can | 1150 | 200 | 0.828 (4.032) | 750 | 200 | 0.765 (5.408) |
| Square | 2600 | 200 | 0.885 (2.086) | 1750 | 200 | 0.799 (3.538) |
| Lift | 300 | 200 | 0.578 (4.196) | 450 | 200 | 0.580 (4.092) |
| Tool Hang | 2650 | 200 | 0.962 (0.563) | 3750 | 200 | 0.932 (1.016) |
| Transport | 2750 | 200 | 0.965 (0.860) | 2800 | 200 | 0.904 (2.356) |
| Block Push | - | - | - | 4800 | 1000 | 0.963 (0.322) |
| Kitchen | - | - | - | 4600 | 566 | 0.704 (14.231) |

Table 2: **Image Obs.** are models conditioned using image observations. **Low Dimensional** are models conditioned on low dimensional states. First number is the average highest similarity with respect to the training trajectories. Second number is the average euclidean distance to the nearest trajectory. Entries with '-' are for tasks with no available checkpoints and therefore could not be evaluated. **Epochs** specify at what epoch the model weights were frozen. All demonstrations and model checkpoints were obtained from Chi et al. (2023).

behavior was consistent across both simulation and real-robot evaluations, reinforcing the robustness of our observations. We acknowledge that extending this analysis to more complex, multi-skill, or longer-horizon tasks, such as sequential, multi-step missions or dexterous hand manipulation, would further strengthen the generality of our conclusions. However, these tasks typically require additional infrastructure and substantially larger demonstration datasets that are difficult to obtain. We therefore leave this analysis for future investigations.

## 4 ACTION LOOKUP TABLE

Building upon these results, we design a lightweight alternative method while still achieving comparable functionality to show the bound of memorization mechanism. Our policy, functioning similarly to a lookup table, uses an image encoder trained with contrastive learning as a hash function to retrieve demonstration trajectories (as shown in Fig. 1). If our hypothesis holds, this method should deliver performance on par with the Diffusion Policy, while also offering more predictable fallback behaviors in the presence of out-of-distribution (OOD) inputs, therefore improving safety and robustness.

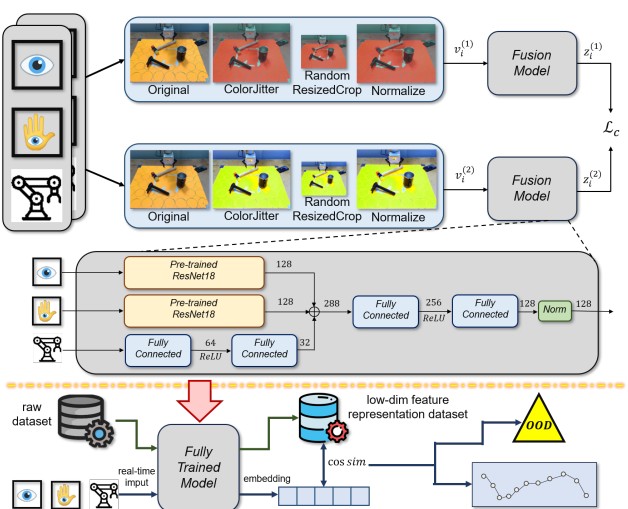

Figure 5: Contrastive training (top, above the yellow dashed line) and inference (bottom) phases of our ALT policy. The inference process has two stages: the green arrows indicate building the ALT latent space with the trained model, while blue arrows represent real-time inference.

### 4.1 METHOD

At each timestep, the dataset includes a first-person end-effector view, a third-person view, and the end-effector pose, denoted as: $D = \{(I_i^h, I_i^t, p_i)\}_{i=1}^N$, where $I_i^h$ and $I_i^t$ are the hand- and third-view images, and $p_i$ is the end-effector position and orientation. We employ a fusion encoder to integrate these inputs into a unified embedding for contrastive learning (Fig. 5). A ResNet-18 He et al. (2016), pretrained on a large-scale dataset, serves as the image encoder backbone.

We adopt a contrastive learning framework to extract robust and discriminative representations from each frame using alignment across multiple modalities. We generate two different augmented views for each sample $d_i = (I_i^h, I_i^t, p_i)$. Specifically, we apply a composed image augmentation pipeline $\mathcal{A}_1$ and $\mathcal{A}_2$ that transforms each input $d_i$ into augmented version views $v_i^{(1)}$ and $v_i^{(2)}$. We feed $v_i^{(1)}$ and $v_i^{(2)}$ into the fusion encoder to obtain their embeddings $z_i^{(1)}$ and $z_i^{(2)}$. These two embeddings form a positive pair, and we train the network using the normalized temperature-scaled cross-entropy (NT-Xent) loss Chen et al. (2020) as our contrastive loss function:

$$\mathcal{L}_c = -\frac{1}{2B} \sum_{i=1}^{B} [\log \frac{\exp\left(\text{sim}(z_i^{(1)}, z_i^{(2)})/\tau\right)}{\sum_{k \neq i}^{2B} \exp\left(\text{sim}(z_i^{(1)}, z_k)/\tau\right)} + (1 \leftrightarrow 2)], \tag{4}$$

Table 3: Experimental Results. InD and OOD denote in- and out-of-distribution cases. The first column reports trajectory retrieval success rate, the second real-robot task success rate, and the remaining columns correspond to Fig. 3. MIT indicates model inference time (s). $w/p$ and $w/op$ refer to using or omitting the end-effector pose as input.

| Methods | Recall | InDs | ID-1 | OOD1 | OOD2 | OOD3 | OOD4 | OOD5 | OOD6 | OOD7 | MIT |
|---------|--------|------|------|------|------|------|------|------|------|------|-----|
| K-D Tree | 100% | 63.3% | ✓ | ✓ | ✓ | ✓ | ✓ | × | ✓ | ✓ | ∼0.09 |
| Diffusion Policy | 100% | 100% | ✓ | ✓ | ✓ | ✓ | × | × | ✓ | ✓ | ∼2.65 |
| ALT w/ p, $\gamma = 0.9$ | 100% | - | ✓ | ✓ | OOD | OOD | OOD | OOD | OOD | ✓ | ∼0.009 |
| ALT w/o p, $\gamma = 0.9$ | 100% | - | ✓ | OOD | OOD | OOD | OOD | OOD | OOD | OOD | ∼0.009 |
| ALT w/o p, $\gamma = 0.75$ | 100% | 100% | ✓ | ✓ | ✓ | ✓ | ✓ | ✓ | OOD | OOD | ∼0.009 |

where $\text{sim}(\cdot, \cdot)$ denotes the cosine similarity, and both inputs are L2-normalized prior to computation. The parameter $\tau$ is the temperature (set as $0.4$ in practice), which controls the sharpness of the similarity distribution, effectively scaling the logits to adjust the contrastive loss sensitivity.

After training, we construct a low-dimensional latent space to enable trajectory matching and prediction (Fig. 5). Each frame in the dataset is encoded by the fusion encoder, and the resulting embeddings, along with trajectory IDs and frame indices, are stored in a database. At inference, the current observation (views and pose) is encoded and compared via cosine similarity against this space. If similarity falls below a threshold $\gamma$, the input is treated as OOD and triggers a safe fallback. Otherwise, the system retrieves the matched trajectory ID and frame index for real-time prediction and policy execution. Although ALT is not a universally scalable solution in its most basic form, as larger datasets can result in higher memory usage and potentially slower lookup, its scalability can be improved through various strategies, including utilizing better data structures, approximate nearest neighbor search, state or action clustering Pertsch et al. (2025); Rothfuss et al. (2018), or through action based contrastive learning Lee et al. (2025).

## 4.2 RESULTS

We evaluate the ALT policy through two experiments: task execution on a real robot under InD conditions, and performance under OOD-Distractors. Comparisons with KD-Tree retrieval and Diffusion Policy validate the explicit action memorization mechanism and the role of representation learning. Results in Table 3 show successful matches (✓) vs. failures (×). Green OOD cases indicate robust OOD detection with correct matches, while red OOD cases denote detected OOD inputs that lead to incorrect trajectories. Our ablations disentangle what enables memorization from how it is implemented.[3] A naïve KD-tree nearest neighbor baseline underperforms, showing that memorization is not just feature proximity—without task-aligned embeddings and temporal alignment, retrieval fails. In contrast, ALT, an explicit action lookup indexed by contrastive embeddings, matches diffusion policy performance on InD and OOD tasks while using far less compute and memory. These results suggest that representation-driven memory is key to closed-loop success: diffusion policy acts as an implicit retrieval system, while ALT exposes the same mechanism explicitly, with lower latency and an interpretable OOD flag.

## 5 CONCLUSION

We investigated why diffusion policies are able to obtain strong performance with few task demonstrations in Visuomotor robot manipulation. We found that much of their closed-loop success in the low-data regime stems from *retrieval-style action memorization*, not sequence-level action generalization. Our comparative study revealed that diffusion policies consistently recall verbatim action sequences from the training data, ACT exhibits interpolation across action segments showing better action generalization, and *GR00T-N1.5* balances action memorization from the fine-tuning phase with robustness likely stemming from the large data volume pretraining of the policy. To probe the limits of simplicity of action memorization, we introduced *ALT*, a minimal, explicit lookup baseline, which reproduces diffusion-level behavior on small datasets while running orders of magnitude faster and offering a simple OOD flag. These findings reframe diffusion policies as powerful action retrieval memory systems that can attain proficiency in a single task, but do not achieve generalizable manipulation skills.

---

[3] Additional details on the encoder architecture and the ablation study of the latent dimension are provided in the Appendix F.

ACKNOWLEDGMENTS

This work was supported in part by NSF grant 2342246, ONR grant N00014-23-1-2354, a gift from Meta, and the ARPA-H HEART project. We are grateful for this support. This research was also partly supported by the Singapore Ministry of Education (MOE), as well as by an NUS Overseas Research Immersion Award.

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

## A    DATA COLLECTION PIPELINE

### A.1    ROBOT ARM DATA COLLECTION

Choosing an efficient, cheap, and safe method for data collection, crucial for robot imitation learning, remains an open problem. One of the most common solutions is to use remote controllers, such as VR, 3D space mouse, or smartphones. However, due to high latency and indirect operation, the data collected in this way is often messy and low-quality, making it difficult to accurately capture human skills. Fully synchronized systems with human operators, such as ALOHA Zhao et al. (2023) and GELLO Wu et al. (2024), can solve this problem by allowing humans to teleoperate the robot in a more intuitive way while tracking the actions of this system in real time. But, these methods require an additional specialized puppeting system, which incurs an additional cost. In comparison, UMI-gripper Chi et al. (2024) is a cheap, intuitive, and robot-agnostic solution for data collection. Yet, it cannot be used in our work as it is incompatible with situations where third-person perspectives are necessary, and limits the robot to a single manipulator that requires an expensive hardware interface. Thus, to collect the necessary data, we utilized a motion capture data collection method, MoDA, to capture high-quality action sequences with low latency.

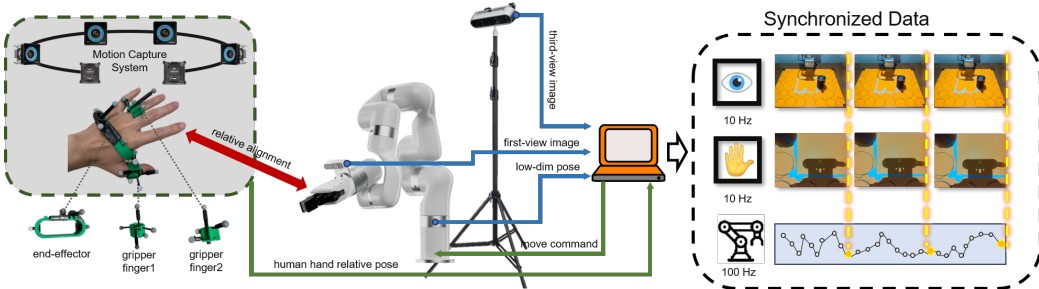

Figure 6: The data collection pipeline of MoDA (Motion-captured Demonstration for Arms). The green arrows indicate the process of aligning the relative positions of human's hand and the robotic arm, and the blue arrows indicate the data of collecting the robotic arm.

### A.2    MOTION-CAPTURED DEMONSTRATION FOR ARMS

Data collection plays a critical role in imitation learning, as the quality and generalizability of the learned policy depends heavily on the fidelity of the demonstrations. In this work, we introduce MoDA (Motion-captured Demonstration for Arms), a streamlined and cost-effective data collection pipeline built upon motion capture (MoCap) systems that are commonly available in robotics laboratories (see Fig. 6). This pipeline provides high-fidelity human demonstrations for the robot, where a human demonstrator performs the cup grasping motion while wearing specialized trackers, and the system translates these motions into corresponding joint targets for a 6-DoF robot arm. MoDA can be extended to any other robot arm system with almost negligible cost, because our data collection pipeline is both task-agnostic and robot-agnostic. To collect the necessary expert training demonstration data, we use an OptiTrack system to track the 6-DoF pose of the human palm in real time and map it directly to the end-effector of a robotic arm. Simultaneously, we estimate the inter-finger distance to control the opening and closing of the gripper, thereby allowing us to signal when to grasp the cup. We then synchronize these actions with the corresponding in-hand and 3rd person camera views. Compared to systems such as ALOHA, which rely on specialized and expensive teleoperation interfaces, our method does not need any active electronics or specialized wearables. Instead, the setup requires only a few 3D-printed brackets to attach passive IR reflective markers to the palm and fingers, making it an extremely low-cost and accessible solution when a MoCap system is already available in the lab. Furthermore, unlike UMI Gripper, which requires direct human interaction during data collection, our setup allows human demonstrators to operate out of frame, thereby ensuring clean third-person video demonstrations. Compared to systems such as ALOHA, which rely on external equipment like teleoperation interfaces or instrumented gloves, our approach avoids the need for expensive or specialized hardware. In contrast to the UMI Gripper generated data, which often involves complex scenes with human demonstrators visibly present in

the frame, our setup enables the collection of clean third-person video demonstrations where human demonstrators are minimally visible. This is particularly beneficial for training diffusion policies, as it minimizes noise and ambiguity in both the action and visual observation spaces, reducing the risk of learning failures due to poor-quality data. In summary, unlike alternative setups that rely on specialized grippers, force sensors, or teleoperation rigs, our system can be assembled in-house with minimal resources and negligible additional expense. Moreover, MoDA is not only task-agnostic, but also robot-agnostic, it does not rely on any specific type or model of robotic arm, making it highly adaptable across different hardware platforms and manipulation scenarios. This flexibility enables seamless integration into a wide range of experimental setups with minimal modification.

## B  EARLY STOPPING EXPERIMENT

To further support our hypothesis, we conducted an early stopping experiment. Early stopping is a common technique used to prevent potential overfitting during training, with the goal of improving a model's generalization ability. In this experiment, we reserved one-third of the dataset as a validation set and used the remaining two-thirds for training. During training, we recorded both the validation loss and the mean squared error (MSE) between the predicted actions and ground-truth actions on the training set. The first metric, validation loss, is used to determine when to stop training, thus preserving the model version with the best generalization. The second metric, actions MSE on training set, is used to monitor the model's performance on the training set. As shown in Fig. 7, although the validation loss reaches its minimum at a certain point, the corresponding action MSE remains high, around 1800. This result indicates that overfitting a diffusion policy model to the training data is a necessary requirement for producing accurate trajectories, as choosing the best model (chosen based on the validation loss) results in a policy that cannot reproduce the correct in-distribution trajectories.

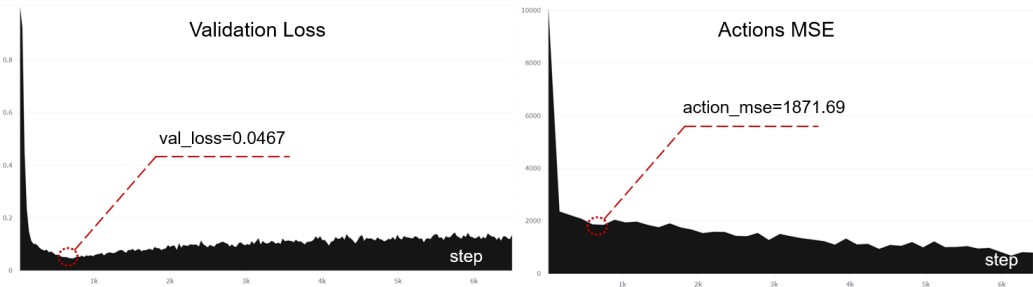

Figure 7: Early Stopping Experiment. The validation loss (left) reaches its minimum around step 650 before beginning to rise, indicating the onset of overfitting. However, at this point, the training action MSE (right) has not yet converged and remains as high as 1800. This suggests that more extensive training is necessary for the Diffusion Policy to output effective actions, even the input is in-distribution.

## C  DIFFUSION MECHANISM ANALYSIS

In this section, we present additional examples to further illustrate the behavior of diffusion models under varying model capacities and dataset scales, as discussed in Section 3.2. Specifically, we examine three additional 2D manifolds: an ellipse 8, a rectangle 9 and a heart shape 10.

Consistent with the observations from Section 3.2, low-capacity models trained on small datasets fail to accurately reconstruct manifolds, often producing noisy or collapsed outputs. Even when trained on large datasets, these models are limited by their representational capacity: simple shapes like ellipses can be approximated reasonably well (albeit still worse than with high-capacity models), more complex structures suffer significant distortion. For example, due to limited model expressiveness, low-capacity models approximate the heart shape as a crude triangle and smooth out the sharp corners of the rectangle.

Memorization behaviors are obvious when high-capacity models are trained on small datasets. Interestingly, although global structure reconstruction fails, local smoothness can still emerge. For

instance, while the overall manifold may not be recovered, segments of an ellipse can still be accurately captured. This suggests that memorization in diffusion models is not absolute: when the training data is locally dense, models may still interpolate between nearby points, preserving some local structure. However, when larger gaps exist between segments, interpolation fails, and the model instead memorizes discrete samples without capturing the broader underlying manifold. When high-capacity models are instead trained on large datasets, all manifolds are accurately reconstructed, almost perfectly matching the true geometry. In practice, though, acquiring such large-scale expert demonstrations that sufficiently cover the state space remains a significant challenge in robotic manipulation tasks.

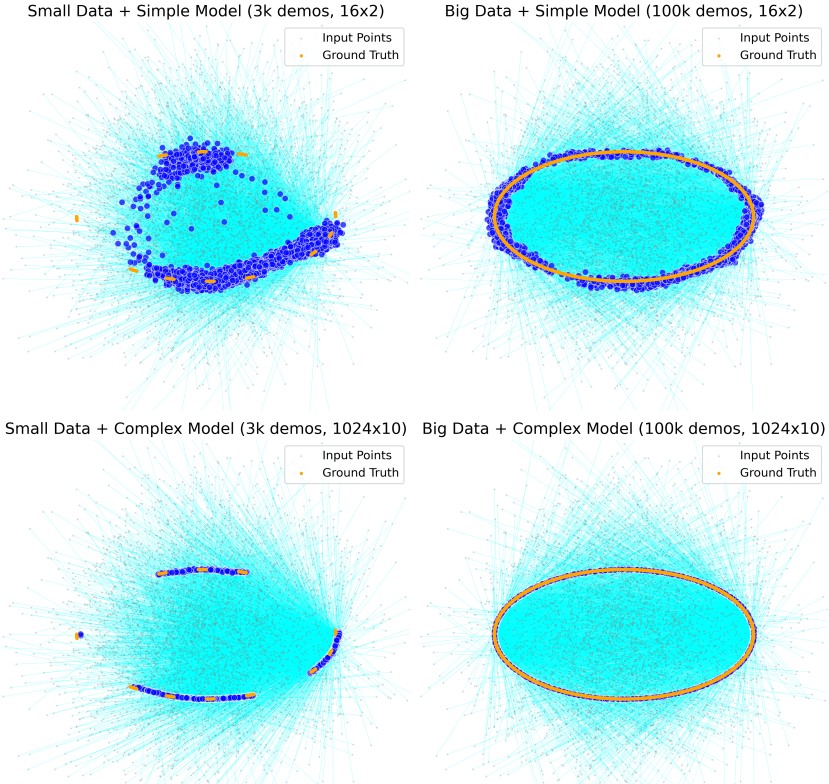

Figure 8: Training a generative model from 2D points on a ellipse-shaped 1D manifold. Orange points indicate training samples, gray points are noisy inputs, blue points are denoised outputs, and cyan lines shows the denoising directions.

## D  ADDITIONAL RESULTS: DIFFUSION POLICY

### D.1  REAL-WORLD

Fig. 11 Shows the trajectory similarity between the predicted trajectory and the nearest ground-truth trajectory for the real-world cup placement experiment. The distance to the nearest neighbor (yellow polyline) is near zero (implying high similarity), while the distance to the second nearest trajectory is substantially larger, resulting in a similarity score that is close to 1 (the blue bar).

Fig. 12 presents additional supplementary visualization of the trajectory matching results under in-distribution conditions. In the left panels, for each case, the first blue bar represents the similarity score of the most closely matched training trajectory, while the second blue bar the similarity score of second closest trajectory. The consistently high top-1 similarity scores, combined with significant gaps to the second closest match, indicate clear and confident retrieval from the training data. As shown in the right panels, the closest trajectory (blue) in the training dataset almost perfectly overlaps with the inference trajectory in all examples, showing that the Diffusion Policy can accurately retrieve the correct demonstration when the input remains within the training distribution. These re-

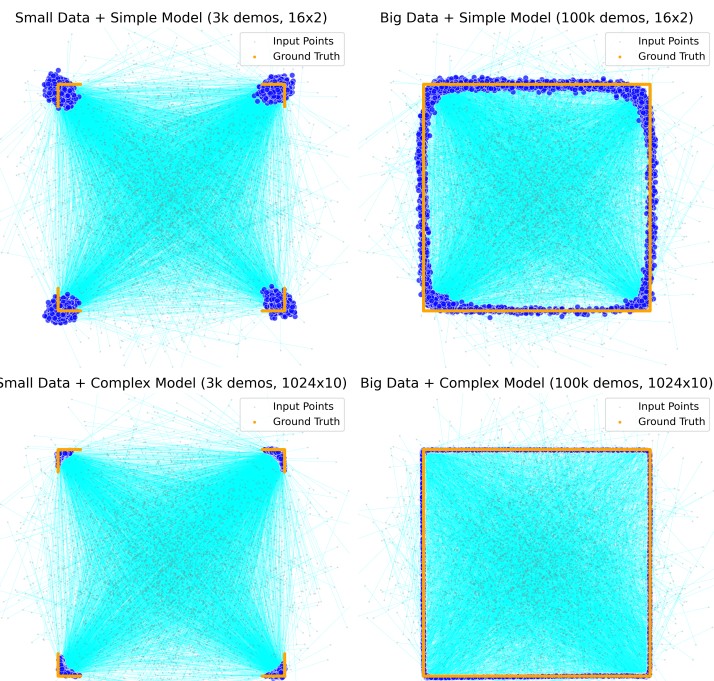

Figure 9: Training a generative model from 2D points on a rectangle-shaped 1D manifold.

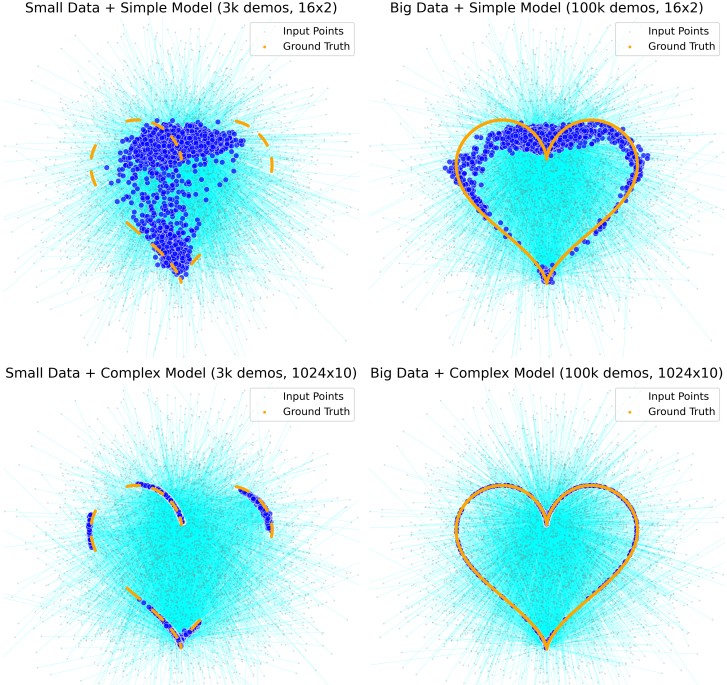

Figure 10: Training a generative model from 2D points on a heart-shaped 1D manifold.

sults strongly support our hypothesis that the Diffusion Policy depends on a memory-based retrieval mechanism to achieve its compelling results. The sharp similarity peaks and trajectory overlaps provide strong evidence that the model is not merely approximating the behavior, but is explicitly recalling memorized training trajectories under in-distribution conditions.

We further analyzed several additional out-of-distribution (OOD) scenarios, including: placing the cup evenly between three or four in-distribution positions (as shown in Fig. 13), gradually moving

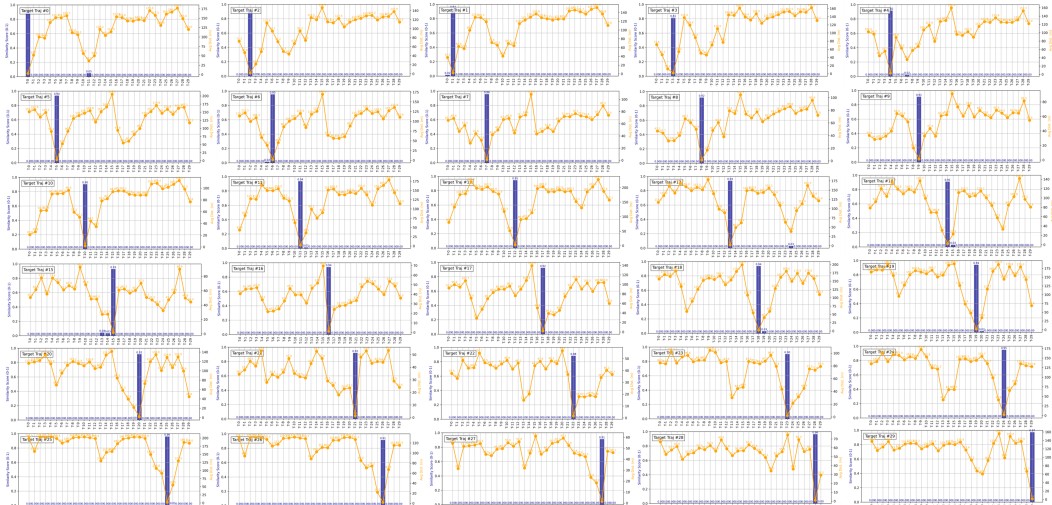

Figure 11: Similarity and distance statistics between inference and training trajectories. Each subplot shows the similarity scores (blue bars) and average distances (orange lines) between the Diffusion Policy inference and training trajectories. The large gap between the closest and second-closest neighbors indicates strong alignment with specific training examples.

the cup out of the field of view from the edge of an in-distribution position (Fig. 14), and slowly transitioning the cup between two distant in-distribution positions (Fig. 15). It is evident that under these OOD conditions, the Diffusion Policy continues to produce trajectories that closely resemble those seen during training. These experimental results provide further support for our core hypothesis regarding the memory-driven behavior of diffusion policies.

## D.2 SIMULATIONS

We also see similar results for the two different simulated manipulation tasks: Can, and Square. Both tasks are drawn from the Robomimic simulation benchmark and contain 196 training trajectories and four validation trajectories. In the Can task, the model is taught to transfer a can from one table to another table while in the "Square" task, the model is trained to place a square ring on a square peg. We observed that these experiments also display action memorization (see Fig. 16 and Fig. 20), as each training observation was found to closely align with the corresponding ground-truth demonstration. We observe similar memorization when examining the diffusion policy's rollouts from observations taken from the validation set (see Figs. 17 through 19, and Figs. 21 through 23) for can and square experiments respectively. Overall, in the Can task, we observed an average similarity score of 0.88 and an average deviation of 2.0 mm. In comparison, in the "Square" task, we observe a similarity score of 0.82 with an average deviation of 4.4 mm.

## E  ADDITIONAL RESULTS: ACT AND GR00T-N1.5

We provide further results for experiments discussed in 3 for both the ACT and GR00T-N1.5 policies on the following scenarios being: (**1:InD-Interpolate**), (**2:OOD-Interpolate**), and (**3:OOD-Extrapolate**).

### E.1  ACT

For OOD-Interpolation in 24 we see trajectories that lie cleanly between training trajectories, demonstrating successful interpolation.

The OOD-Extrapolation case in 25 often extrapolates in the wrong direction leading to incorrect trajectories.

| Encoder-Dim | InDs | InD-1 | OOD1 | OOD2 | OOD3 | OOD4 | OOD5 | OOD6 | OOD7 |
|---|---|---|---|---|---|---|---|---|---|
| ResNet-64 | 100% | ✓ | ✓ | ✓ | ✓ | ✗ | ✗ | ✓ | ✓ |
| ResNet-128 | 100% | ✓ | ✓ | ✓ | ✓ | ✗ | ✗ | ✓ | ✓ |
| ResNet-256 | 100% | ✓ | ✓ | ✓ | ✓ | ✗ | ✗ | ✓ | ✓ |
| SimpleCNN-64 | 19.35% | ✗ | ✗ | ✗ | ✗ | ✗ | ✗ | ✗ | ✗ |
| SimpleCNN-128 | 22.58% | ✗ | ✗ | ✗ | ✗ | ✗ | ✗ | ✗ | ✗ |
| SimpleCNN-256 | 22.58% | ✗ | ✗ | ✗ | ✗ | ✗ | ✗ | ✗ | ✗ |
| ViT-64 | 12.9% | ✓ | ✓ | ✓ | ✓ | ✓ | ✓ | ✓ | ✗ |
| ViT-128 | 12.9% | ✓ | ✓ | ✓ | ✓ | ✓ | ✓ | ✓ | ✓ |
| ViT-256 | 16.13% | ✗ | ✗ | ✗ | ✗ | ✗ | ✗ | ✗ | ✗ |
| CLIP-64 | 22.58% | ✓ | ✓ | ✓ | ✓ | ✗ | ✗ | ✗ | ✗ |
| CLIP-128 | 100% | ✓ | ✓ | ✓ | ✓ | ✗ | ✗ | ✗ | ✗ |
| CLIP-256 | 51.61% | ✓ | ✓ | ✗ | ✓ | ✓ | ✓ | ✗ | ✗ |
| Swin-64 | 77.42% | ✓ | ✓ | ✓ | ✓ | ✗ | ✗ | ✓ | ✓ |
| Swin-128 | 83.87% | ✓ | ✓ | ✓ | ✓ | ✓ | ✗ | ✓ | ✓ |
| Swin-256 | 90.32% | ✓ | ✓ | ✓ | ✓ | ✗ | ✗ | ✓ | ✓ |

Table 4: Performance of ALT under Different Encoder Architectures and Feature Dimensions.

## E.2 GR00T-N1.5

From the InD-Interpolation of 26 we see high similarity scores to a single trained trajectory over many samples. This demonstrates high action-memorization while sometimes interpolating.

The OOD-Interpolation in 27 show some successful interpolation, but we see lower action-memorization through the increased number of higher similarity score candidate trajectories.

For OOD-Extrapolation, 28, highly OOD inputs can lead to incorrect extrapolations. There is some minor-action memorization for successful extrapolations.

## F  ENCODER ARCHITECTURE AND LATENT DIMENSION

To further understand the role of representation quality in ALT, we evaluated the system under five different encoder architectures—ResNet-18, SimpleCNN, ViT, CLIP, and Swin—while varying their feature dimensions. The results in Table 4 reveal several consistent trends. First, ResNet-18 achieves uniformly high performance across all dimensions, reflecting the stability and strong inductive biases of convolutional networks, which are well suited for manipulation-oriented visual inputs. In contrast, SimpleCNN and ViT perform significantly worse: the former lacks sufficient representational capacity, while the latter requires large-scale pretraining and offers limited spatial inductive bias, making its embeddings unstable in our domain. CLIP produces strong features but is highly sensitive to output dimensionality—dimensions that are too small under-represent the feature manifold, whereas excessively large dimensions distort it, leading to degraded contrastive alignment. Finally, Swin exhibits a clear performance improvement as dimensionality increases, consistent with its hierarchical, local-window design that combines CNN-like locality with transformer expressiveness. Overall, these results highlight that encoder choice and representation quality have a substantial impact on ALT performance, confirming that representation quality is a critical factor for contrastive training.

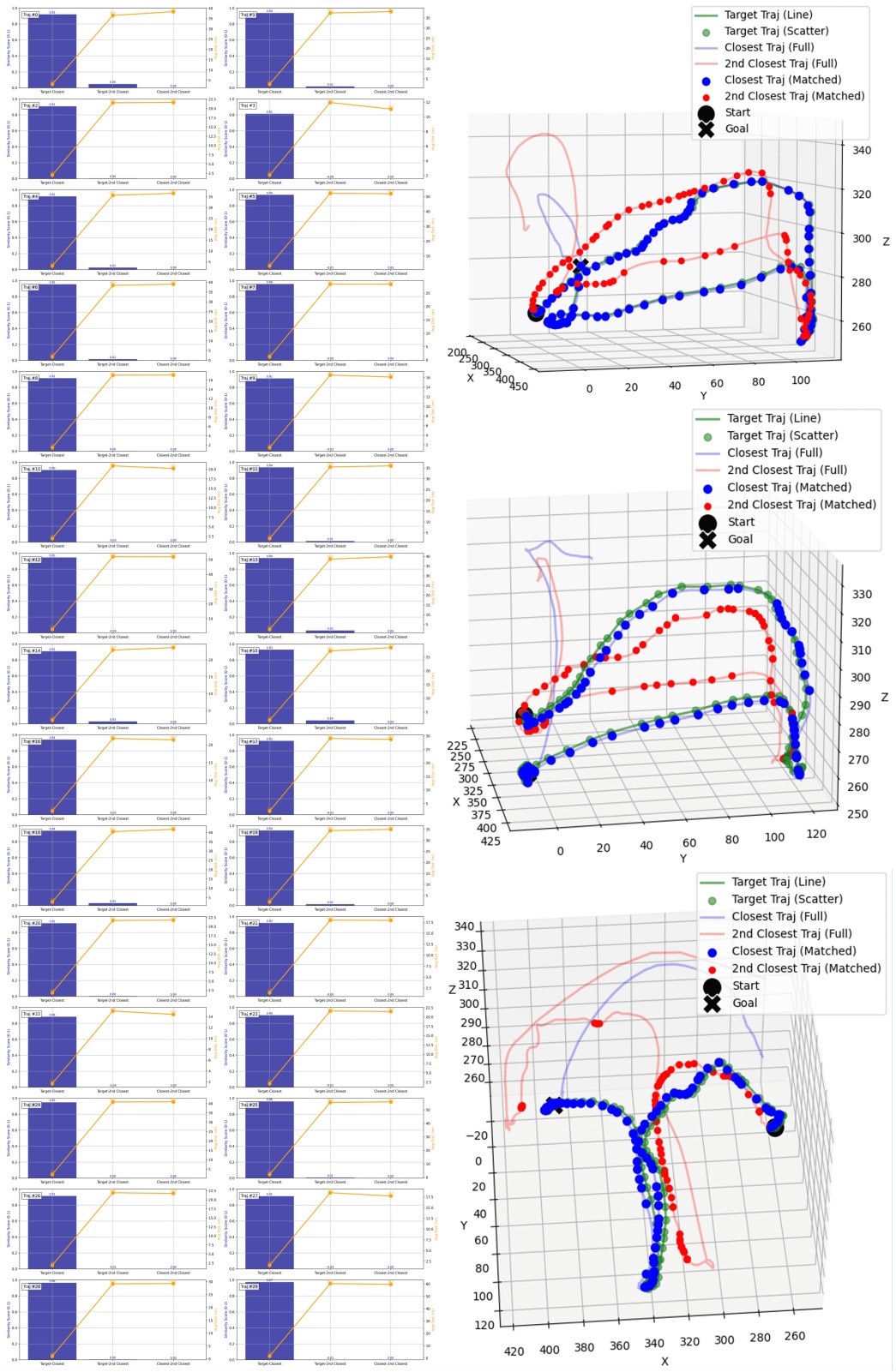

Figure 12: The Diffusion Policy inference result analysis under in-distribution conditions. On the left, each subplot shows the similarity distribution between the query inference trajectory and all stored trajectories in the database. On the right, the three figures provide 3D visualizations of representative matching cases. The green line represents the inference trajectory, while the blue and red dots show the closest and second-closest trajectories retrieved from the training set, respectively.

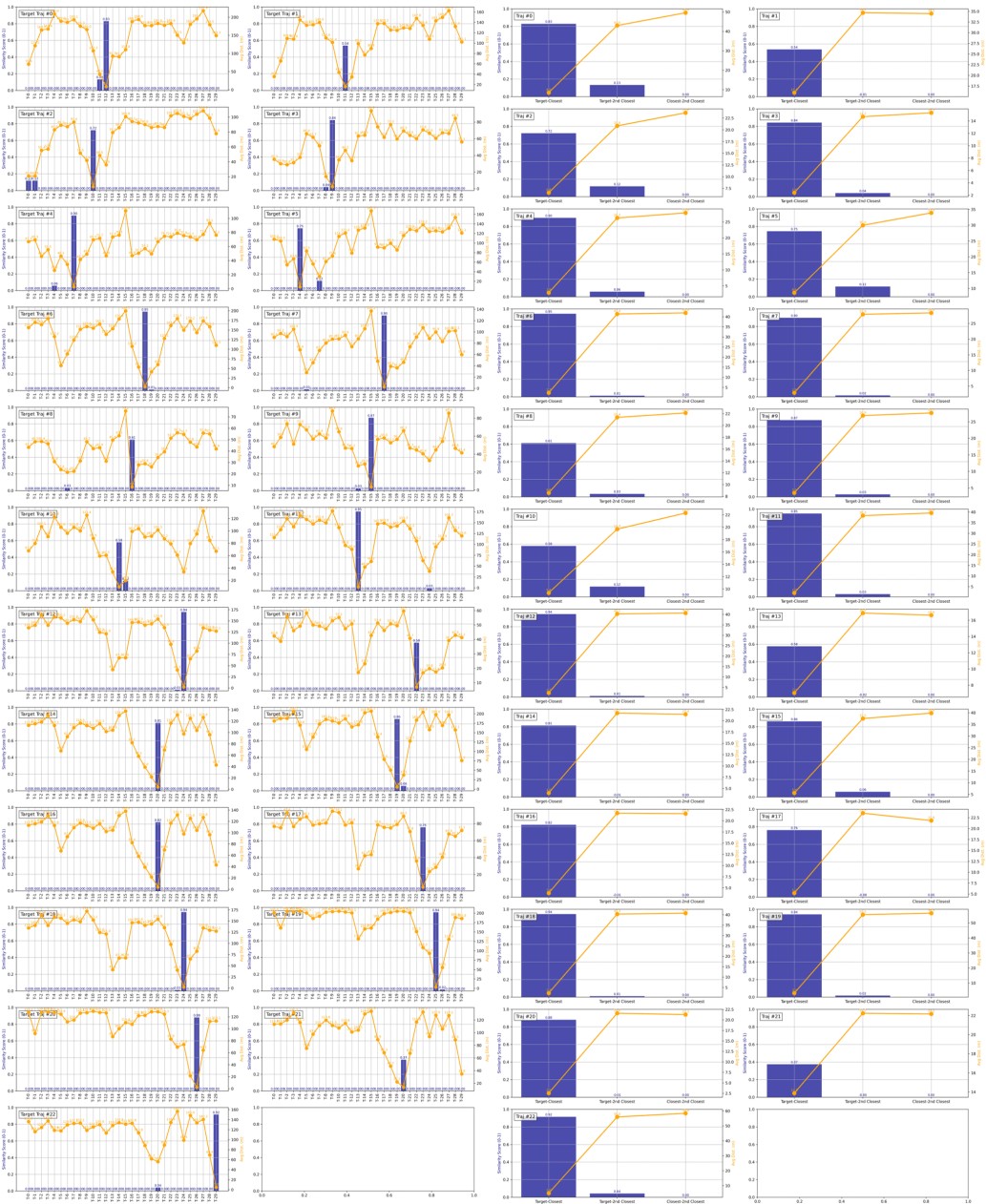

Figure 13: Inference trajectory analysis when the cup is placed between multiple In-distribution positions.

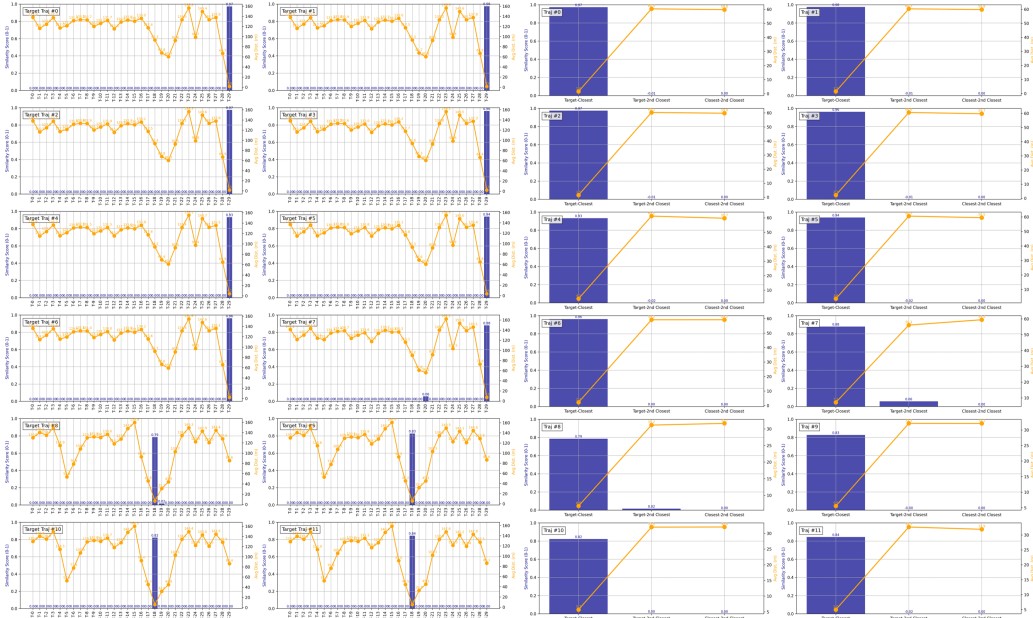

Figure 14: Inference trajectory analysis when the cup is gradually moved out of view from an in-distribution boundary.

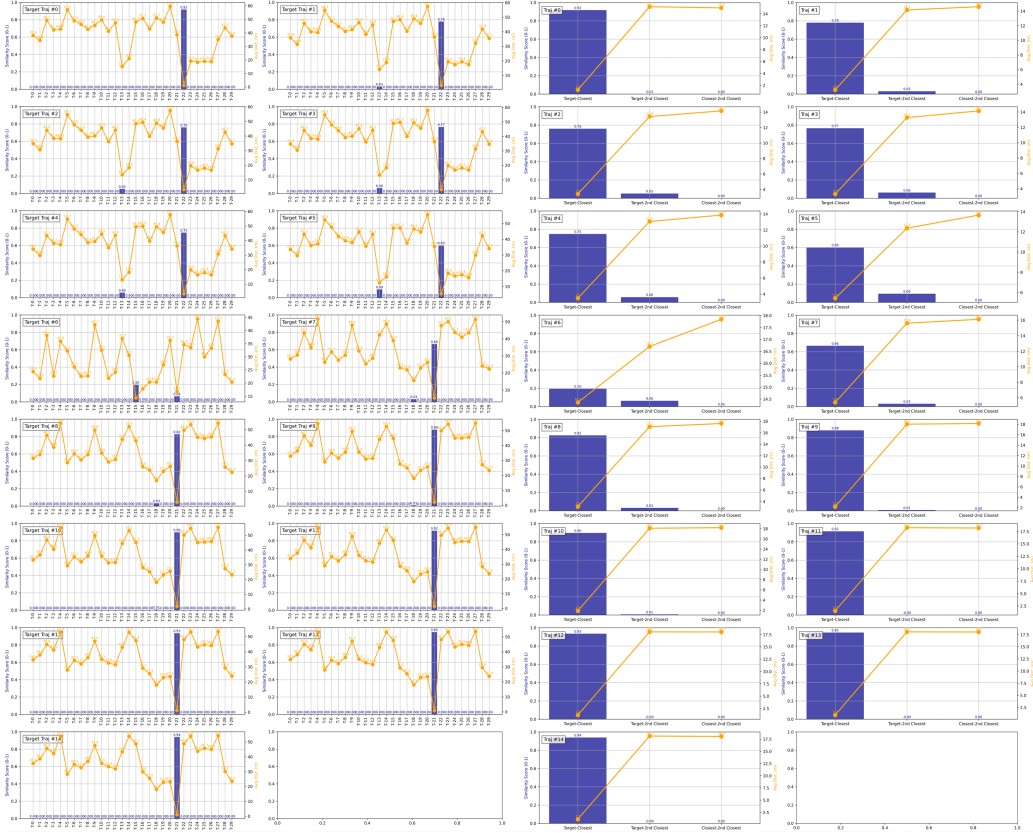

Figure 15: Inference trajectory analysis when the cup is gradually moved between two distant in-distribution positions.

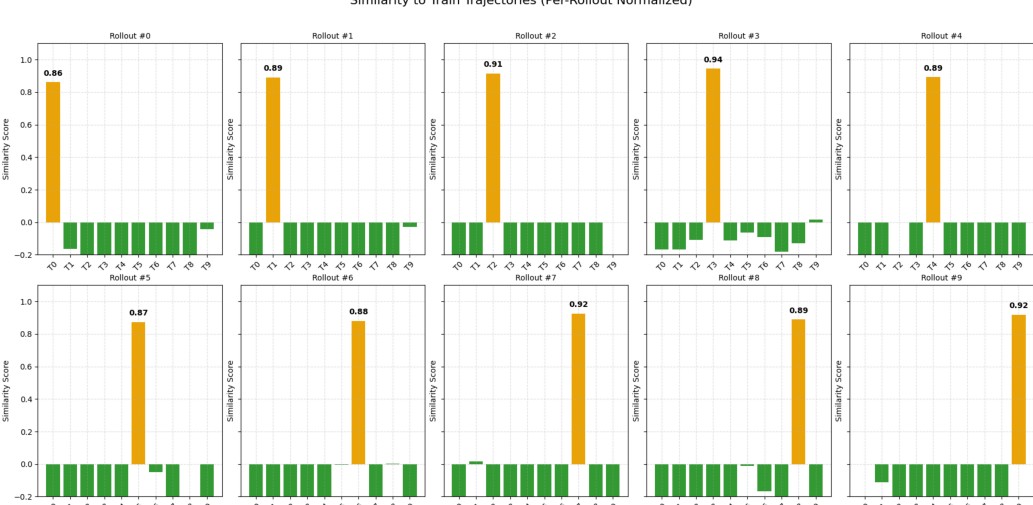

Figure 16: A subset of the total analysis done for the roll-outs of a Diffusion Policy model from the training set with respect to all ground-truth training trajectories. The model was trained on the "Can" pickup benchmark.

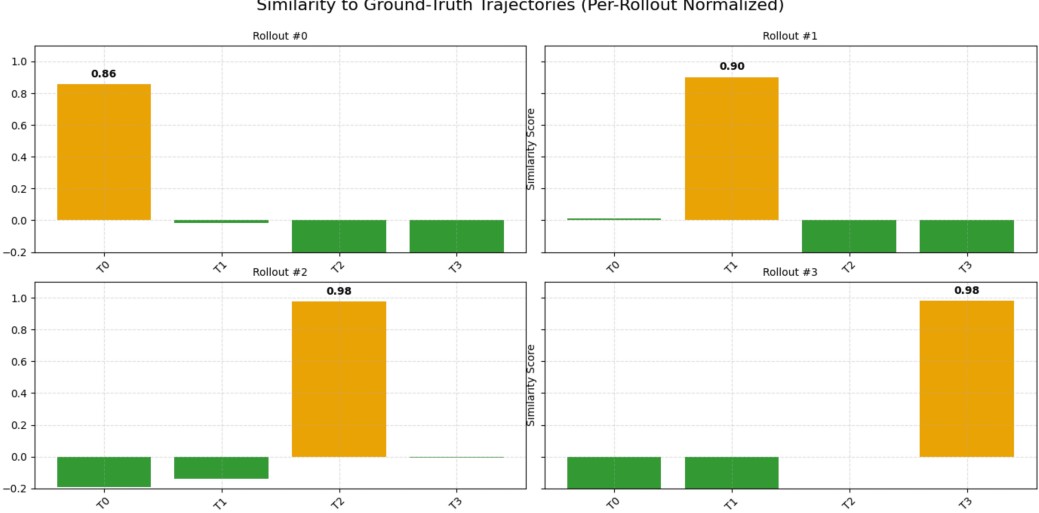

Figure 17: Analysis of a Diffusion Policy model roll-outs on the validation set with respect to all ground-truth validation trajectories. The model was trained on the "Can" pickup benchmark.

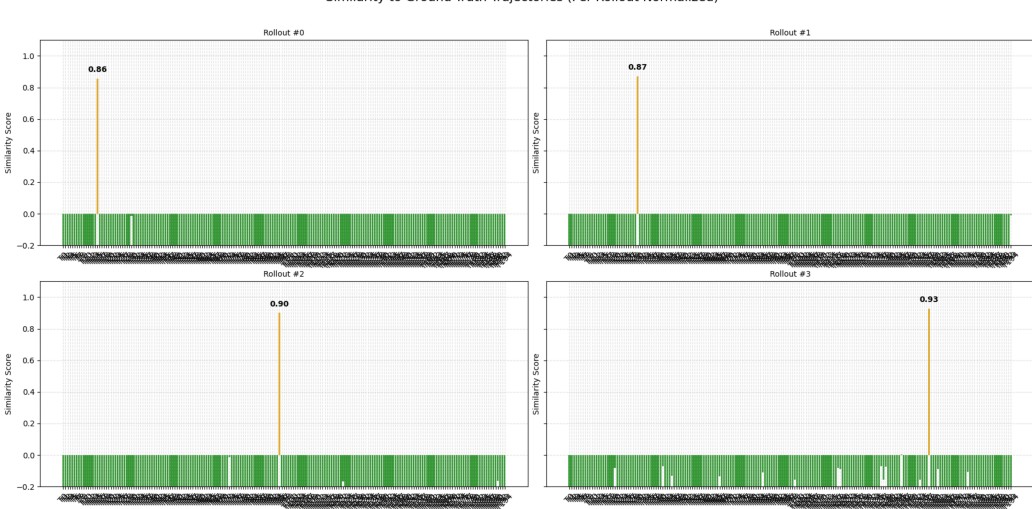

Figure 18: Analysis of a Diffusion Policy model roll-outs on the validation set with respect to both the ground-truth training and validation trajectories. The model was trained on the "Can" pickup benchmark.

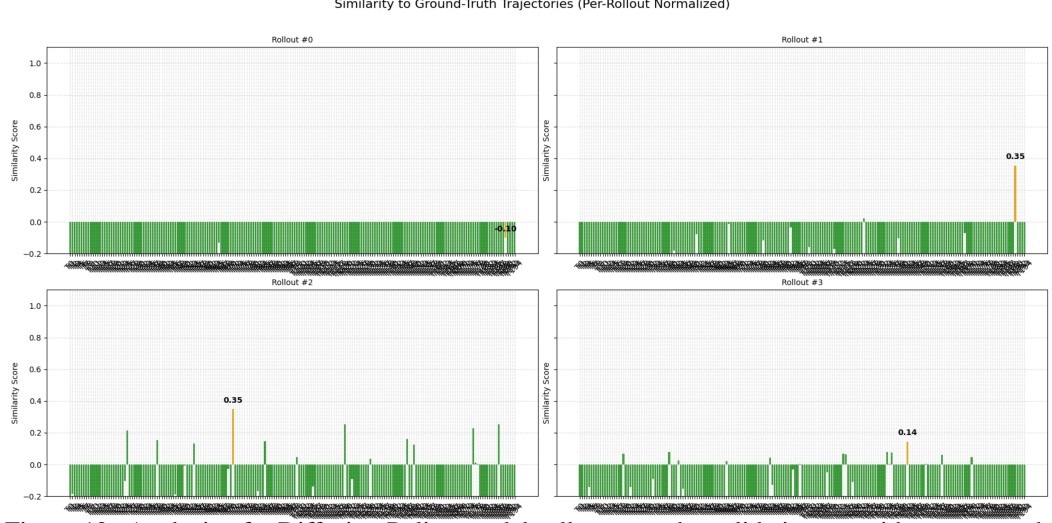

Figure 19: Analysis of a Diffusion Policy model roll-outs on the validation set with respect to the ground-truth training trajectories. The model was trained on the "Can" pickup benchmark.

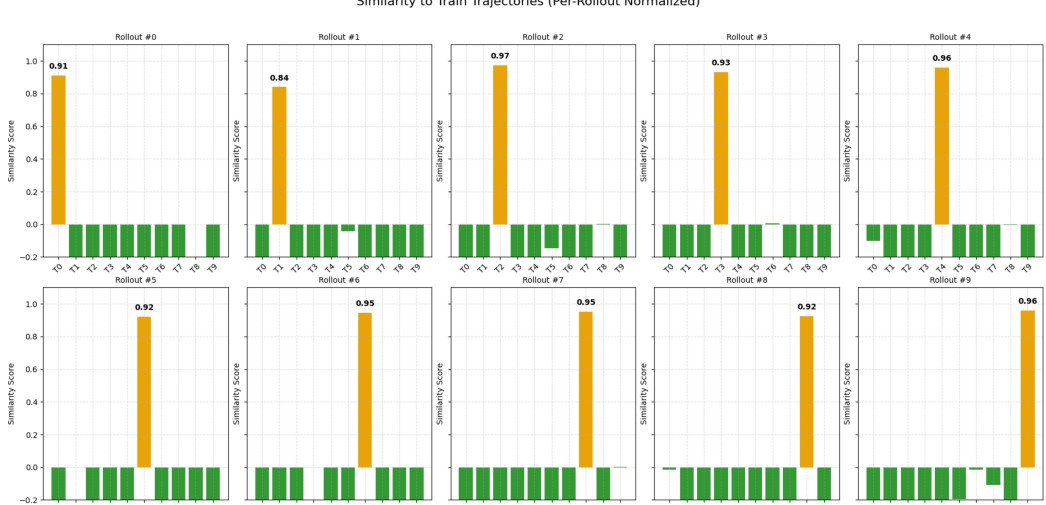

Figure 20: A subset of the total analysis done for the roll-outs of a Diffusion Policy model from the training set with respect to all ground-truth training trajectories. The model was trained on the "Square" pickup benchmark.

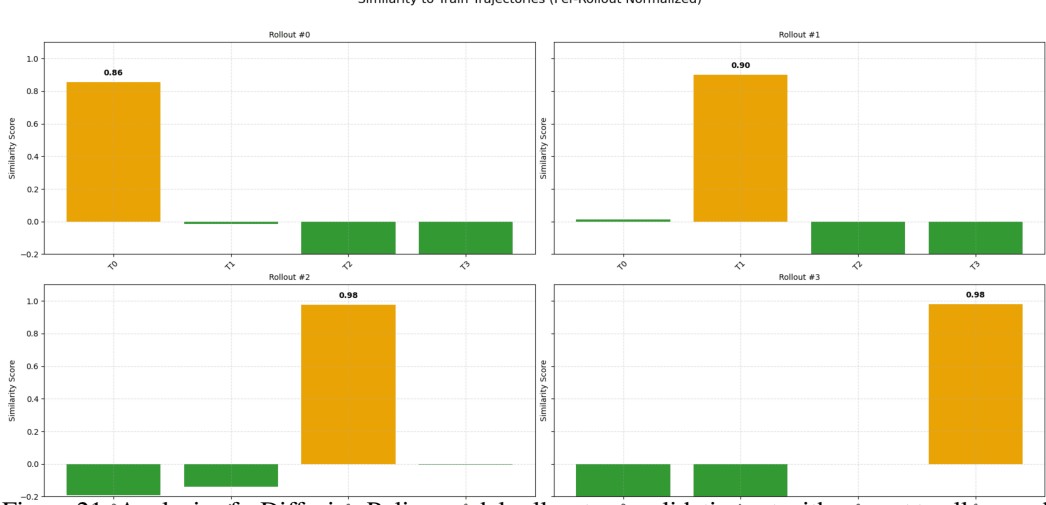

Figure 21: Analysis of a Diffusion Policy model roll-outs on validation set with respect to all ground-truth validation trajectories. The model was trained on the "Square" benchmark.

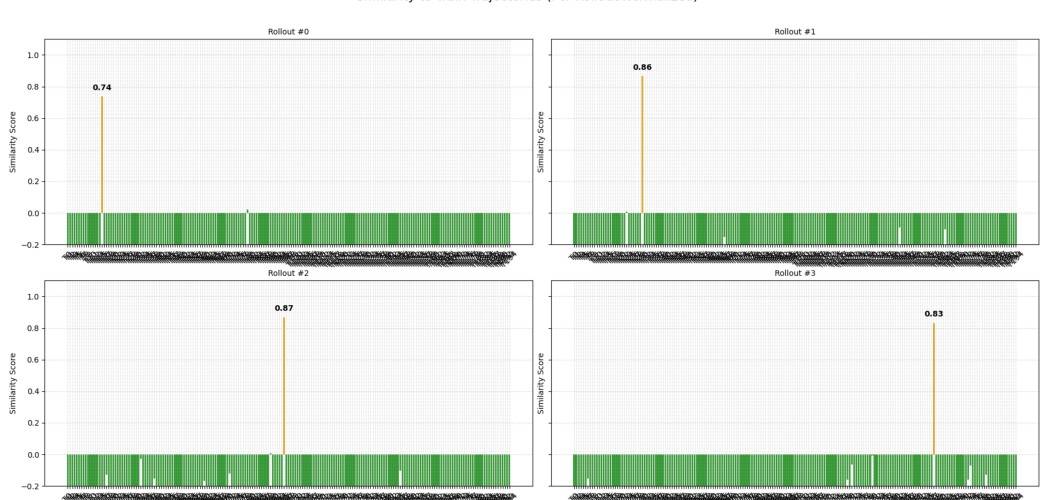

Figure 22: Analysis of a Diffusion Policy model roll-outs on validation set with respect to both the ground-truth training and validation trajectories. The model was trained on the "Square" pickup benchmark.

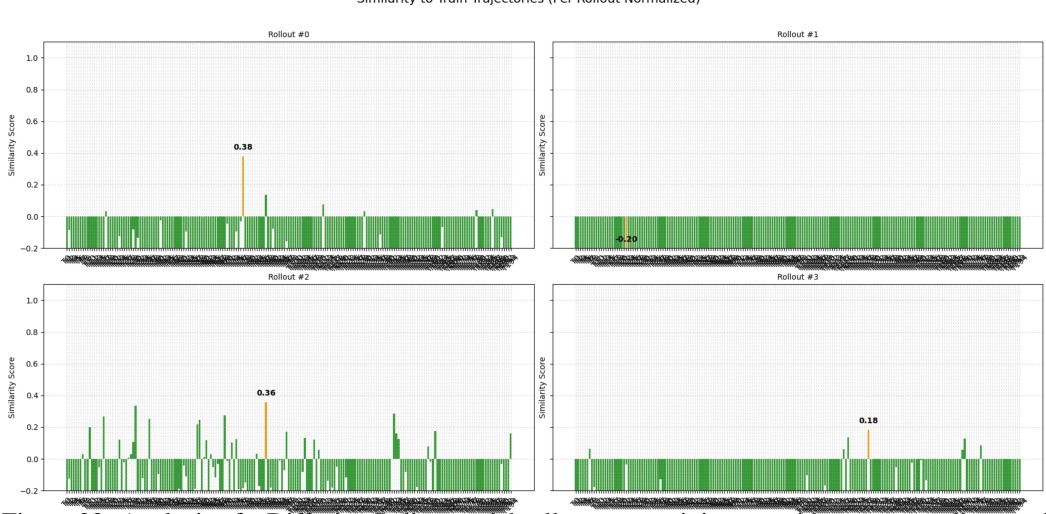

Figure 23: Analysis of a Diffusion Policy model roll-outs on training set with respect to all ground-truth training trajectories. The model was trained on the "Square" benchmark.

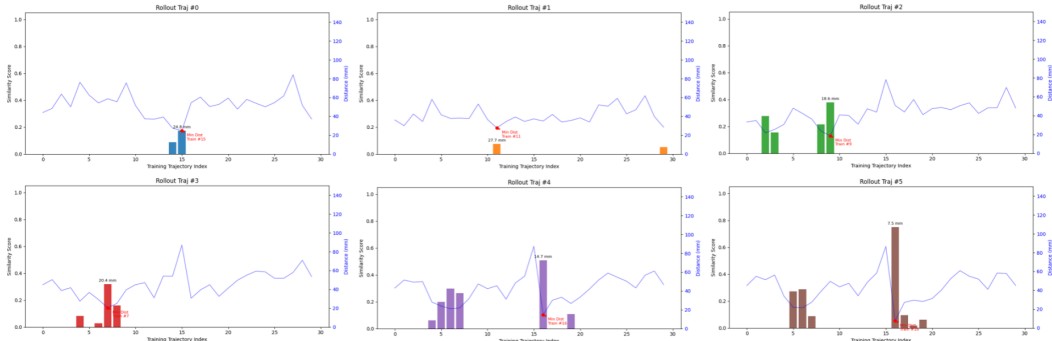

Figure 24: ACT Out of Distribution Interpolation Distance and Similarity Scores Across Six Roll-outs.

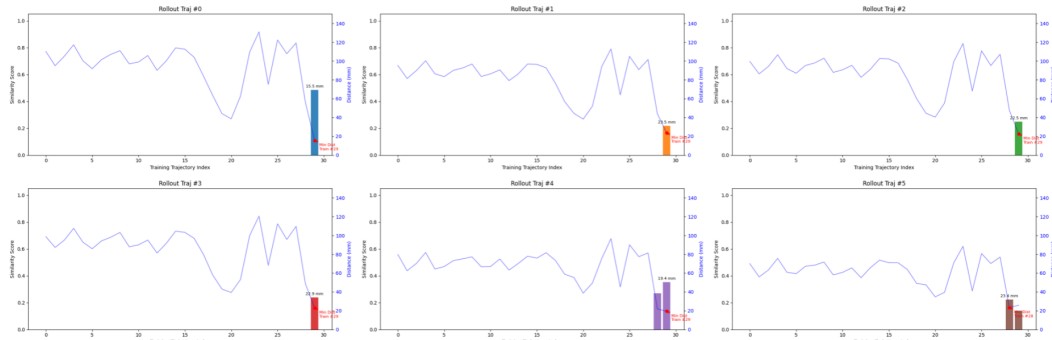

Figure 25: ACT Out of Distribution Extrapolation Distance and Similarity Scores Across Six Rollouts.

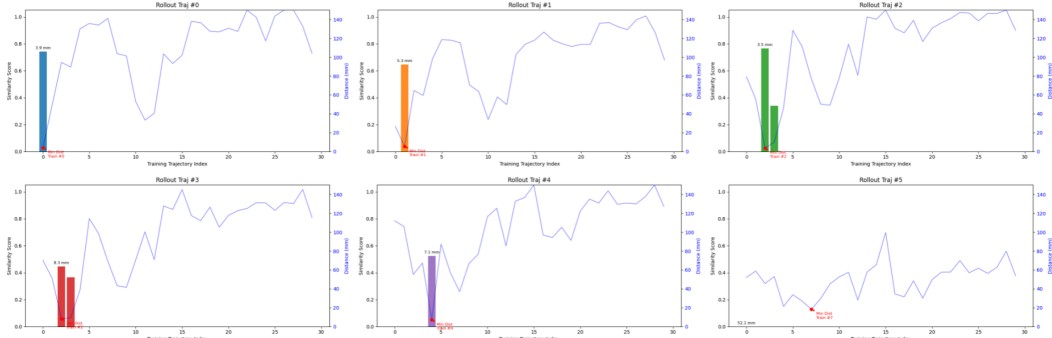

Figure 26: GR00T-N1.5 In Distribution Interpolation Distance and Similarity Scores Across Six Rollouts.

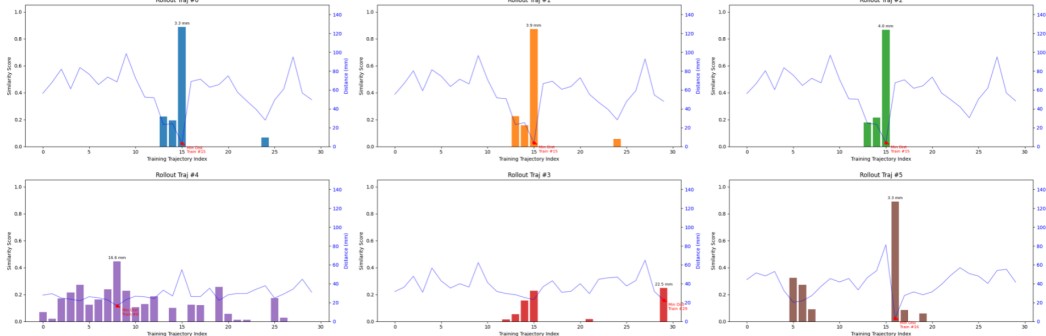

Figure 27: GR00T-N1.5 Out of Distribution Interpolation Distance and Similarity Scores Across Six Rollouts.

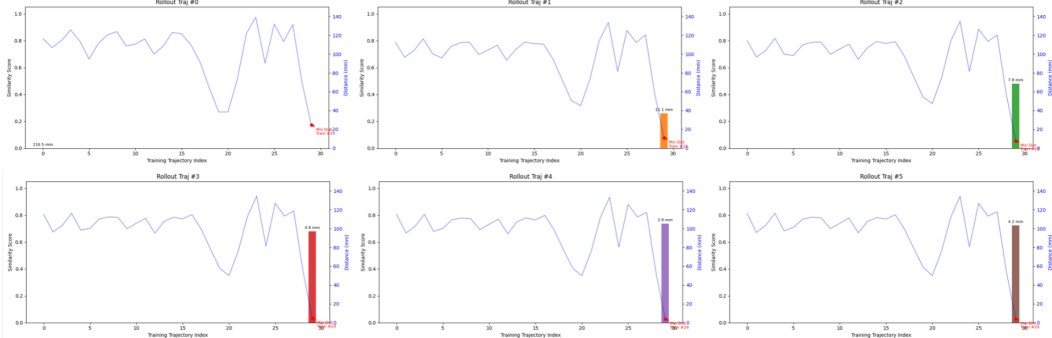

Figure 28: GR00T-N1.5 Out of Distribution Extrapolation Distance and Similarity Scores Across Six Rollouts.

