# OpenReview forum: "Demystifying Robot Diffusion Policies: Action Memorization and a Simple Lookup Table Alternative"
_ICLR.cc/2026/Conference — ICLR 2026 Poster_

### Official Review · Reviewer_kN3g · 2025-10-31

**Soundness:** 4
**Presentation:** 3
**Contribution:** 4
**Rating:** 8
**Confidence:** 4

**Summary:**

The work investigates the effectiveness of Diffusion Policy (DP) for visuomotor robot manipulation, proving that their performance in low-data regimes stems from action memorization rather than any sort of generalization. Empirical evidence demonstrates that the DP tends to recall action sequences from its training data, even when faced with highly OOD inputs, providing robustness in those scenarios. The authors compare this behavior with other architectures like ACT, which shows action interpolation but lacks robustness, and GR00T N1.5, which shows generalization due to pre-training. To validate their hypothesis on DP, the researchers introduce a simpler, faster alternative called the Action Lookup Table (ALT) policy, which explicitly relies on nearest neighbour lookup in an embedding space, to match DP's performance while also providing OOD detection. The (image & joint pose) encoder that creates the embedding is trained via contrastive learning.

Basically, this work does a great job reframing DP, in data-sparse regimes, as a memory retrieval system.

**Strengths:**

1. Well explained idea. Strong paper to put forward the point of memorization.
2. The example of grasping the cup, along with the memory-audit metric S, is simple and effective.

**Weaknesses:**

1. Sec 3.3 & Fig 3. More OOD variations could be added. Specifically, what if you change the cup object itself?
2. Fig 4 is quite confusing at first sight. I had to read it again to understand that the lines indicate distance scores. Considering removing information to make it clearer. Do you need these absolute distance scores, when you already have a similarity metric? in that metric, you can see the one-hot like nature of DP anyway.
3. ALT does not attempt to interpolate between training data actions. However, this may be a design choice by the authors.

**Questions:**

1. "The diffusion model seems to revert to one or two fallback action sequences when presented with OOD images." Based on FIg-4, it seems like that trajecotry is traj4 ? Why do you think there is a single fallback demo? And why is that the chosen traj? Is it the one earliest in the training? What characteristic does traj4 have that sets it apart from the others. A deeper discussion into might provide insight into the importance of each demo in the training pipeline.

2.Fig 5: Please improve visibility of the lines in this figure. The green and blue arrows are too similar, especially for such a small figure. Consider higher-contrast colours, or use dots and dashes.

---

> ### Author Response · Authors · 2025-11-21
> **Response to Reviewer kN3g**
>
> We greatly appreciate the reviewer’s positive evaluation and insightful suggestions. Please find our point-by-point responses below.
>
> Response to Weaknesses #1:
>
> We thank the reviewer for the suggestion of introducing additional OOD experiments. Changing the cup object would be a significant OOD change that could reasonably be considered a different task depending on how different the new object is. For example, it is well known that simply changing the color of a target object can result in task failure in visuomotor policies [1]. Furthermore, as demonstrated by our other OOD scenarios in which other objects are ignored, changing the cup shape would also likely result in the cup being ignored as well thereby resulting in task failure. Thus, this OOD scenario would likely result in all methods failing. Moreover, this is a visually significant change that would be easily detected by our OOD detector due to the differences in the image pixels with respect to our training demonstrations.
>
> [1] 3D Diffusion Policy: Generalizable Visuomotor Policy Learning via Simple 3D Representations, Ze et. al., 2024
>
> Response to Weaknesses #2:
>
> We apologize for the confusion and will make the figure clearer in the final submission. While the similarity score allows us to know which two demonstrations are similar, it does not describe how close they are spatially. Conceivably it possible to have two trajectories look similar but be shifted in space far away from each other, and thus being two distinct and unrelated demonstrations. Furthermore, in situations where a policy interpolates between two training examples, we would expect to find the resulting rollout to be situated in between both training demonstrations in addition to having a partial similarity to both. Therefore, the absolute comparison allows us to distinguish between interpolation, memorization, and similar but distinct trajectories.
>
> Response to Weaknesses #3:
>
> The reviewer is correct that ALT does not interpolate between training actions. We will clarify this in the paper. ALT operates by selecting and replaying segments from the nearest demonstration trajectory, rather than blending or averaging actions from multiple demos. Interpolating between demonstrations (especially in high-dimensional visuomotor space) could lead to unnatural or unsafe actions, so we prioritized fidelity to real examples. We will add a sentence to emphasize that this non-interpolative property of ALT is by design, aimed at ensuring safety and reliability. Future work could be pushing towards interpolation in latent space.
>
> Response to Questions #1:
>
> In our experiments we found that the trajectory chosen as the fall back depends both on the task and the demonstrations used for that task. We hypothesize that this likely due to a combination of how the optimizer is initialized, how complicated the task is, and on how similar demonstrations are to each other, which can affect what initial action to take. We agree with the reviewer that this is an interesting avenue for future exploration.
>
> Response to Questions #2:
>
> We apologize to the reviewer for the confusion. We will improve the readability of the figure for the final submission.

---

### Official Review · Reviewer_p3Mt · 2025-11-01

**Soundness:** 3
**Presentation:** 4
**Contribution:** 3
**Rating:** 8
**Confidence:** 4

**Summary:**

The authors hyphothsize that visuomotor policies, e.g., diffusion policies (DP) memorize the image-action pairs instead of learn to generlize on them. To verify this, they conduct a systematic emprical study including analysis and experiments ranging from toy data to real-robot data. Moreover, they evaluate on three representative policy architectures, namely DP, Action Chunking with Transformers (ACT), and GR00T and found that they exhibit different behaviors. Based on this, a simple alternative called action lookup table implemented as observation encoder for action retrievial is introduced and requires less memory while providing greater run-time efficiency and OOD monitoring capability.

**Strengths:**

- An interesting and comprehensive emprical study for a highly relevant but under-explored problem of current visuomotor imitation learning framwork. I believe that this investigation is inspiring for both researchers and practioners.

- The presentation of this submission is clear, well-organized and easy-to-follow along with illusrative figures and visualization;

- The toy experiments and visuliazation provide a great example for the reader to under the problem at the beginning;

- The analysis of results across three architectures is presented clear and concise, while also making connection to the manifold attraction effects.

- The experiment is extensive from the perspective of hyphothesis validation. Morever, the experimental design (ID and OOD setups), policy architectures and evaluation metrics selection further consolidate the strictness of this empirical investigation.

- This study is practical-relevant and highly informative for practioners, because in the real-world scenarios a large amount of robot data may be costly to collect.

**Weaknesses:**

- The current study and robot-related experiments are mainly done under the low-data regime. It remains unclear what implications this study can bring for the visuomotor imitation learning at larger scale. It would be valuable to provide insights (discussion or experimental study) about the validity for a large scale of data, e.g., 200, 2000 or 20k demostrations are provided.

- Though the presention is clear and great, I would still suggest to shorten the part of analysis for the three architectures or to present the difference or similarites of them in a more visual way. Then more space can be left for a slightly proper experiment part, including setup introduction and more results.

- In the experiment, the performance of ACT and GR00T for reference is missing; In table 2., success rates on OOD scenarios would be nice to have to investigate the expected behaviors described in section 3.3.

- Regarding the proposed ALT, besides the improved computation efficiency, its OOD detection capability is highly desirable for robustness enhancement. It would be highly valuable to perform more experiments for this part to understand how effective it can be. For example, a comparison with other density-based approaches such as normalizing flows, which can be trained on the latent features.

[1] Feng, J., Lee, J., Geisler, S., Günnemann, S., & Triebel, R. (2023, December). Topology-Matching Normalizing Flows for Out-of-Distribution Detection in Robot Learning. In Conference on Robot Learning (pp. 3214-3241). PMLR.

**Questions:**

- The influence of the feature dimensionality of ALT on the task performance seems relevant. Is this a core desgin question?

---

> ### Author Response · Authors · 2025-11-21
> **Response to Reviewer p3Mt**
>
> Thank you for your positive evaluation and thoughtful comments. We appreciate the reviewer’s careful reading of our work and address each of the raised points in detail below.
>
> Response to Weaknesses #1:
>
> We thank the reviewer for raising the question of scaling to larger demonstration datasets. While our experiments were conducted in a low-data regime, we also briefly explored larger data regimes (3k and 100k demonstrations) in manuscript's Figure 2. As expected, larger datasets result in better generalization. Performing a similar scale for real world experiments is out of scope for our paper due to the difficulty in collecting the necessary demonstrations. Nevertheless, we expect to see a similar trend to the one we observed in simulation.
>
> Response to Weaknesses #2:
>
> We appreciate the suggestion to streamline the architecture analysis. In the revision, we will condense the description of these architectures.
>
> Response to Weaknesses #3:
>
> We thank the reviewer for pointing out this oversight on our part. We are actively working on rectifying this omission for the final submission.  It takes time to bring up these policies on our hardware, so these reults are not ready for this round of rebuttal, but we expect them to be ready for the final round of rebuttal.  As shown in manuscript's Figure 3, we found that both ACT and GR00T are robust to visual distractors so long as the intended target object is in view. Therefore, we fully expect ACT and GR00T to have equivalent performance with DP in the Recall, InD, and OOD cases 1 through 3. Furthermore, We expect GR00T to perform better than or equivalent to DP in the OOD cases 4 and 5 since it was able to produce trajectories with higher similarity to the ground truth paths when presented with a similar OOD event. Similarly, we expect ACT to perform equivalent to or worse than DP due to it being more sensitive to visual distractors than GR00T or DP. In addition, similar to DP, neither ACT nor GR00T have a method to detect when an OOD situation has occurred. Thus, we expect similar results for them as well.
>
> Response to Weaknesses #4:
>
> We fully agree that ALT's OOD detection capability deserves a separate research effort to compare its effectiveness to the reviewer's reference [1] and other recently introduced OOD detection methods.  Given scope, space, and time limitations, we leave this for future work as this was not the main focus of this paper.  However, intuitively, in cases where the OOD scenarios are expected to be easily recognizable by the vision encoder, such as in the case of observations containing new visual distractors or unexpected camera changes, we expect our OOD method will perform well. On the other hand, in situations where the OOD observations are due to dynamics difference (different friction or material properties) which are not reflected in the vision encoder, we anticipate that our method will struggle, as would any method based on visual OOD queues.
>
> Response to Questions #1:
>
> We acknowledge that the feature dimensionality of ALT is an important design choice. In our current implementation, we used a fixed latent feature dimension (we will explicitly state the value used). We will discuss its influence on performance. Intuitively, this dimension controls the expressive power of ALT’s latent space: too low might under-represent demonstration features, while too high could introduce noise or redundancy (as well as computationally and memory complexity).
>
> We experimented with different feature dimensions—64, 128, and 256—using our previous architecture. With ResNet, the performance remained largely unchanged, likely due to its stable embedding characteristics. In contrast, when using CLIP as the encoder, the choice of feature dimension had a pronounced effect on model performance. As shown in Table 1, selecting an appropriate dimension is essential: dimensions that are too small or too large can degrade performance to varying degrees.
>
> **Table 1: The impact of different feature dims on different encoder architectures**
> | Encoder-Dim   | InDs    | InD-1 | OOD1 | OOD2 | OOD3 | OOD4 | OOD5 | OOD6 | OOD7 |
> |---------------|---------|-------|------|------|------|------|------|------|------|
> | ResNet-64     | 100%    | ✓     | ✓    | ✓    | ✓    | ✗    | ✗    | ✓    | ✓    |
> | ResNet-128    | 100%    | ✓     | ✓    | ✓    | ✓    | ✗    | ✗    | ✓    | ✓    |
> | ResNet-256    | 100%    | ✓     | ✓    | ✓    | ✓    | ✗    | ✗    | ✓    | ✓    |
> | CLIP-64       | 22.58%  | ✓     | ✓    | ✓    | ✓    | ✗    | ✗    | ✗    | ✗    |
> | CLIP-128      | 100%    | ✓     | ✓    | ✓    | ✓    | ✗    | ✗    | ✗    | ✗    |
> | CLIP-256      | 51.61%  | ✓     | ✓    | ✗    | ✓    | ✓    | ✓    | ✗    | ✗    |

---

### Official Review · Reviewer_zjbw · 2025-11-02

**Soundness:** 3
**Presentation:** 3
**Contribution:** 2
**Rating:** 4
**Confidence:** 4

**Summary:**

The main claim of the paper is that diffusion policy in low-data regime memorize demonstration actions during training and looks it up during inference, and proposes Action Lookup Table policy. Experiments with small datasets show that diffusion outputs memorized action chunks even when given out of distribution inputs. ALT also introduces out-of-distribution detection and stochastic kNN sampling, and observes that overfitting can be beneficial in manipulation setups.

**Strengths:**

- Clear claim with sound evidence from sim and real experiments.
- ALT performs well in the low data regime and is orders of magnitude faster than diffusion / ACT.
- Out-of-distribution experiments show that diffusion policy replays action chunks from the training set.

**Weaknesses:**

- My main concern is novelty. The idea that kNN lookup as a policy class on contrastively learned features works well in the low-data regime has been well-established in literature [1], and it is also well-known in the field that overfitting can help with downstream rollouts in behavioral cloning.
- Limited experiments. Main claims are mostly validated with cup grasping experiments, with low effective state and action space dims. It would be good to see whether the diffusion memorization holds in higher dim action / state spaces (e.g. dexterous manipulation for higher dim actions, or benchmarks with more objects like LIBERO), as well as a sweep over dataset scale.

[1] The Surprising Effectiveness of Representation Learning for Visual Imitation. Pari et al., 2021.

**Questions:**

- Does this claim hold on a higher dim action space, and environments with multiple objects? Does it hold on a larger scale dataset?
- Ablation: how much does the representation learning matter?

---

> ### Author Response · Authors · 2025-11-21
> **Response to Reviewer zjbw - Part 1**
>
> Thank you for your review and constructive feedback. We appreciate your comments and address each point below.
>
> Response to Weaknesses #1:
>
> Thank you for bringing up this point. We recognize that the individual components of our approach (using a contrastively learned representation and a kNN-style policy) have been studied in prior research. Indeed, previous studies in visual imitation learning have reported that a good feature representation combined with a non-parametric policy can be surprisingly effective with small datasets. We do not claim to have invented the idea of nearest-neighbor policies. Instead, our contribution is in applying and analyzing this idea in the context of Diffusion Policies, and thereby demystifying why Diffusion Policies perform so well with limited data. While it may seem unsurprising that a high-capacity model will overfit to a small dataset, our work is to systematically demonstrate that Diffusion Policy’s ``overfitting” takes the form of almost verbatim action recall. We provide extensive empirical evidence, e.g., showing that even under extreme visual perturbations the Diffusion model outputs an action sequence from its training set, to substantiate this interpretation. By formulating the ALT baseline, we support this insight by providing an easy-to-understand alternative that mirrors Diffusion Policy’s behavior. We believe this is novel and useful from an analysis standpoint. Rather than proposing an algorithmic innovation for raw performance gain, we are contributing an explanation and a new perspective. We also show practical insights: for instance, ALT’s ability to flag OOD inputs and its speed advantage are concrete benefits that fell out of our analysis.
>
>
> Response to Weaknesses #2:
>
> We acknowledge the scope of our experiments was constrained, and we understand the desire to see the analysis extended to more diverse settings. In our submission, we focused on a few tasks that are representative of common benchmarks in robot imitation learning: lifting a cup (real robot pick-and-place task) and two simulation tasks (''Can'' and ''Square'' from the Robomimic suite). These tasks have relatively low-dimensional action spaces and a single object of interest. We chose them because they allowed us to meticulously study how each policy behaves under controlled variations. This level of analysis would have been difficult on a complex multi-object or dexterous hand task without an enormous data collection effort. Beyond these tasks we also explored other simulated tasks from the robomimic dataset, which are presented in Table 1. It is also worth noting that our task count and complexity are in line with other works in this area (see the original Diffusion Policy paper and ACT). We agree that pushing to higher-dimensional or more complex tasks is important for future research. In fact, part of our motivation was to identify whether Diffusion Policy’s strategy would scale or change in those more challenging scenarios. We expect that with more data, Diffusion Policy might gradually transition from pure memorization to more interpolation or even generalization. ALT would also benefit from more data (more entries to draw from), but beyond a point, a learned model might become more parameter-efficient than storing massive memories. In essence, our study captures one end of the spectrum (small data); exploring the continuum towards big data is something we plan to pursue. We will add a discussion to acknowledge this limitation and to make clear that our conclusions pertain to the small-data regime. We appreciate the reviewer’s suggestion to broaden the experimental evidence and will highlight the need for such experiments going forward.
>
> **Table 1**: First number is the average highest similarity while the second number in the parentheses is the euclidean distance. '-' means the checkpoints were unavailable.
> | Task        |  Epochs | # Demos | Image Obs. | Epochs | # Demos | Low-Dim Obs. |
> |-------------|-------------------|---------|---------------------------|----------------|---------|------------------------|
> | Can         | 1150              | 200     | 0.828 (4.032)             | 750            | 200     | 0.765 (5.408)          |
> | Square      | 2600              | 200     | 0.885 (2.086)             | 1750           | 200     | 0.799 (3.538)          |
> | Lift        | 300               | 200     | 0.578 (4.196)             | 450            | 200     | 0.580 (4.092)          |
> | Tool Hang   | 2650              | 200     | 0.962 (0.563)             | 3750           | 200     | 0.9324 (1.016)         |
> | Transport   | 2750              | 200     | 0.965 (0.860)             | 2800           | 200     | 0.904 (2.356)          |
> | Block Push  | -                 | -       | -                         | 4800           | 1000    | 0.963 (0.322)          |
> | Kitchen     | -                 | -       | -                         | 4600           | 566     | 0.704 (14.231)         |

---

> > ### Author Response · Authors · 2025-11-21
> > **Response to Reviewer zjbw - Part 2**
> >
> > Response to Questions #1:
> >
> > This comment overlaps with the previous one, and we echo our response here. Since we did not test, for example, a 24-DoF dexterous hand or a multi-object manipulation scenario in this work, we cannot claim with certainty that diffusion policies would rely on memorization to the same extent in those cases. Our findings primarily cover the low-data, single-task setting. Specifically, we use the same datasets as those used by methods such as diffusion policy in different simulation environments at Robomimic. In real robot experiments, we use datasets ranging from tens to over a hundred demonstrations, consistent with the dataset sizes used by well-known methods such as diffusion policy and action chunking transformer (ACT). We will explicitly note in the paper that our claims are scoped to the conditions tested.
> >
> > Response to Questions #2:
> >
> > The role of representation learning is absolutely important, and we’re glad the reviewer raised this point. We anticipated that if ALT was to match a learned policy, it would need a strong representation, and our results confirmed this. As detailed in the paper, we performed an ablation by using a naive nearest-neighbor approach (which we implemented with a KD-tree for efficiency) without the benefit of our task-specific contrastive features. This baseline served as a proxy for ”no/poor representation learning.“ The outcome was clear: the naive approach underperformed. Conversely, when we use the well-trained contrastive encoder, ALT’s retrievals align with the correct actions, and its success rate matches that of Diffusion Policy on our tasks. We will make sure to emphasize in the paper that the contrastive representation is a crucial component enabling ALT’s performance.
> >
> > In response to the reviewer’s question, the ablation clearly shows that representation learning is crucial for ALT. As shown in Table 2, switching only the encoder—while keeping the contrastive objective, training pipeline, and downstream architecture identical—leads to dramatic performance differences (from 12.9\% with ViT to 100\% with ResNet and CLIP). This large variance demonstrates that ALT’s performance is fundamentally tied to the quality and inductive biases of the learned visual representations, confirming that representation learning plays a central role.
> >
> > **Table 2: ALT performance under different encoder architectures.**
> > | Encoder   | InDs    | InD-1 | OOD1 | OOD2 | OOD3 | OOD4 | OOD5 | OOD6 | OOD7 |
> > |---------------|---------|-------|------|------|------|------|------|------|------|
> > | ResNet    | 100%    | ✓     | ✓    | ✓    | ✓    | ✗    | ✗    | ✓    | ✓    |
> > | SimpleCNN | 22.58%  | ✗     | ✗    | ✗    | ✗    | ✗    | ✗    | ✗    | ✗    |
> > | ViT       | 12.9%   | ✓     | ✓    | ✓    | ✓    | ✓    | ✓    | ✓    | ✓    |
> > | CLIP      | 100%    | ✓     | ✓    | ✓    | ✓    | ✗    | ✗    | ✗    | ✗    |
> > | Swin      | 83.87%  | ✓     | ✓    | ✓    | ✓    | ✓    | ✗    | ✓    | ✓    |

---

### Official Review · Reviewer_aGTM · 2025-11-06

**Soundness:** 3
**Presentation:** 2
**Contribution:** 2
**Rating:** 2
**Confidence:** 4

**Summary:**

This paper investigates why diffusion policies perform remarkably well in robot manipulation despite being trained on very few demonstrations. The authors hypothesize that diffusion policies do not truly generalize actions, but instead memorize and retrieve action sequences-similar to a nearest-neighbor lookup table-based on the closest-matching observation in latent space. Through real and simulated pick-place experiments, they show that Diffusion Policy consistently reproduces training trajectories even under out-of-distribution inputs, while ACT interpolates actions but fails in OOD settings, and GR00T combines some generalization with robustness due to large-scale pretraining. Motivated by these findings, they introduce a simple Action Lookup Table (ALT) approach using contrastive visual embeddings, which matches Diffusion Policy performance in low-data regimes while being faster and explicitly flagging OOD cases. The key takeaway is that the strength of Diffusion Policy in low-demo robot learning stems largely from memorization, not generalization, and that simple retrieval-style methods can provide similar performance with far lower cost and greater interpretability.

**Strengths:**

1. The paper is well-organized, progressing logically from hypothesis to experimental validation and practical takeaway.
2. The paper performs a sound analysis of Diffusion Policy and other single-task imitation learning methods that are known to perform well in the low-data regime. Diffusion Policy is indeed recommended to train to overfitting, and the paper tries to make the effects of that trend very clear.
3. The authors design structured experiments (In-distribution, OOD-interpolate, OOD-extrapolate, and OOD-distractor) to dissect memorization versus generalization behavior, making the conclusions grounded and interpretable.
4. The introduction of the Action Lookup Table (ALT) policy provides a transparent, computationally efficient alternative that matches diffusion policy performance while being ~300x faster and offering explicit OOD detection.

**Weaknesses:**

1. While I appreciate the analysis, the paper, in my opinion, does not provide a novel viable solution. It just presents a simple alternative to Diffusion Policy, which is not scalable to something that current large behavior models aim for. While I agree that Diffusion Policy is prone to overfitting, then so is any high-capacity supervised learning model trained on less data. Diffusion Policy did not just present one specific architecture or loss function, but provided a starting point for future approaches, which might be able to have less overfitting and more generalization while maintaining the multi-modality and dexterity that Diffusion Policy has. As such, I did not see anything in this paper which I did not expect from low-data supervised learning models, but it does not mean that researchers should replace them with non-scalable methods such as presented in this paper.
2. The experiments in this paper are primarily conducted on simple manipulation tasks (e.g., cup grasping, pick-and-place), making it unclear whether the findings generalize to more complex, multi-skill or long-horizon robotic tasks.

**Questions:**

1. How much does ALT generalize, or is capable of generalizing to novel object placement arrangements, lighting, etc.? How does lookup table size scale with generalization capabilities?
2. As the number of demonstrations grows, how does ALT’s retrieval latency, memory footprint, and OOD detection accuracy scale? Would a nearest-neighbor search remain feasible for larger datasets or multi-task scenarios?
3. How would the ALT policy perform on tasks that require longer-horizon reasoning, contact-rich dynamics, or multi-step planning? Would its lookup-based retrieval break down when task stages are less visually correlated?
4. The paper uses a contrastive encoder to map images to latent representations. How sensitive is ALT’s performance to the choice of encoder architecture or the quality of contrastive learning?

---

> ### Author Response · Authors · 2025-11-21
> **Reponse to Reviewer aGTM - Part 1**
>
> We thank the reviewer for carefully reading our manuscript and for providing constructive feedback. We appreciate the concerns raised and have addressed each point in detail below.
>
> Our response to Weaknesses #1:
>
> Thank you for this feedback. We would like to clarify that our work is intended as an analysis of Diffusion Policy behavior rather than a proposal of a superior replacement. We agree that Diffusion Policy is a powerful approach – our findings do not imply it is ``bad.” Instead, we show that in a low-data regime, Diffusion Policy’s impressive performance stems largely from leveraging training data (essentially recalling seen action sequences) rather than true generalization. This observation is **not meant to undercut** Diffusion Policy’s value, but to provide an objective explanation for why it works so well with limited data. In addition, we would like to **emphasize** that the data collected and used in our experiments is **comparable** to the amount suggested by the original paper.
>
> In the meantime, we acknowledge that ALT in its basic form is not designed to scale up to very large action spaces or multi-task scenarios out-of-the-box. It was chosen as a simple, transparent baseline to contrast with Diffusion Policy – not as a universally scalable solution. If one wants to extend memory-based policies like ALT, there are promising strategies (e.g. using approximate nearest-neighbor search for speed, clustering similar states/actions using learned compression schemes [1] to compress the search space, or by using a latent action representation [2]) that could mitigate scalability issues. [3] presents a policy named CLASS that is conceptually dual to ALT—training the model through action-based contrastive learning rather than observation-based learning. This approach further highlights the strong potential of combining representation learning with action-sequence retrieval. Therefore, we believe that ALT and our analysis about these polices offers the community a new insight: for diffusion policies trained with limited data but large model capacity, the unique advantages of memory can be leveraged to design more effective diffusion-policy implementations.
>
> In summary, our goal is to offer a constructive analysis on Diffusion Policy's excellent performance and relate it to observations in the research community that diffusion policies tend to memorize its demonstrations. We use ALT to demonstrate this principle in a pared-down way. We present this insight to inspire future Diffusion Policy variants that maintain their multi-modal and high-dimensional strengths while improving generalization – a direction we fully agree is important.
>
> [1] FAST: Efficient Action Tokenization for Vision-Language-Action Models, Pertsch et. al., 2025
>
> [2] Deep Episodic Memory: Encoding, Recalling, and Predicting Episodic Experiences for Robot Action Execution, Rothfuss et. al., 2018
>
> [3] Lee, S.W., Kang, X., Yang, B.Y. and Kuo, Y.L., 2025, October. Class: Contrastive learning via action sequence supervision for robot manipulation. In Conference on Robot Learning (pp. 4743-4766). PMLR.

---

> ### Author Response · Authors · 2025-11-21
> **Reponse to Reviewer aGTM - Part 2**
>
> Our response to Weaknesses #2:
>
> Thank you for raising this concern. Indeed, our evaluation focused on three representative manipulation tasks rather than an extensive multi-task suite. These include a real-world pick-and-place task (lifting and placing a cup) and two simulated tasks from the Robomimic benchmark (a “Can” picking task and a “Square” block manipulation task). We chose this scope to allow careful, systematic analysis of policy behaviors in the low-data regime. This approach is in line with many prior works in imitation learning and diffusion policies (including the original Diffusion Policy work), which often evaluate on a handful of similarly scoped tasks. Focusing on a limited task set enabled us to perform detailed experiments (including fine-grained out-of-distribution tests) to validate our hypothesis. Although not currently shown in our paper, we have also explored several other tasks from the Robomimic dataset, which are shown in Table 1. These experiments were performed using the training demos, evaluation trajectories and stored checkpoints provided by the original Diffusion Policy paper. As shown in Table 1, with the exception of the lift task, the rollout trajectories from diffusion policies have a high degree of similarity with their training demonstrations, which again indicates model memorization rather than generalization from the training dataset. In the case of the lift task, we hypothesize that lower similarity was because the model was trained for much fewer epochs and because it was the simplest of all the tasks performed. We will make sure to add these results in the appendix as well.
>
> The insights we report – e.g. the tendency of Diffusion Policy to retrieve familiar actions – were consistent across all our tasks (simulation and real robot). We agree that demonstrating the phenomena on more complex, multi-skill or longer-horizon tasks (such as sequential multi-step missions or dexterous hand manipulation) would strengthen generality. However, such tasks typically require much larger demonstration datasets or additional infrastructure. Investigating those was beyond our current scope, especially since we wanted to stay in the small-data regime where memorization effects are easier to observe. We will clarify in the paper that our conclusions are drawn for the types of tasks tested, and we’ll emphasize that extending the analysis to more complex domains is a direction for future work.
>
> **Table 1**: First number is the average highest similarity with respect to the training trajectories while the second number in the parentheses is the euclidean distance to the nearest trajectory. Entries with '-' are for tasks where the checkpoints were unavailable and so could not be evaluated at this time. **Image Obs.** are models conditioned using image observations while **Low Dimensional** are models conditioned on low dimensional states. **Epochs** specify at what epoch the model weights were frozen. All demos and model checkpoints were obtained from the Diffusion Policy Paper.
> | Task        |  Epochs | # Demos | Image Obs. | Epochs | # Demos | Low-Dimensional |
> |-------------|-------------------|---------|---------------------------|----------------|---------|------------------------|
> | Can         | 1150              | 200     | 0.828 (4.032)             | 750            | 200     | 0.765 (5.408)          |
> | Square      | 2600              | 200     | 0.885 (2.086)             | 1750           | 200     | 0.799 (3.538)          |
> | Lift        | 300               | 200     | 0.578 (4.196)             | 450            | 200     | 0.580 (4.092)          |
> | Tool Hang   | 2650              | 200     | 0.962 (0.563)             | 3750           | 200     | 0.9324 (1.016)         |
> | Transport   | 2750              | 200     | 0.965 (0.860)             | 2800           | 200     | 0.904 (2.356)          |
> | Block Push  | -                 | -       | -                         | 4800           | 1000    | 0.963 (0.322)          |
> | Kitchen     | -                 | -       | -                         | 4600           | 566     | 0.704 (14.231)         |

---

> > ### Author Response · Authors · 2025-11-21
> > **Reponse to Reviewer aGTM - Part 3**
> >
> > Our response to Questions #1:
> >
> > This is a good question. By design, ALT's behavior is to match the closest known observation and replay its corresponding action. If the test scenario is a variation of something in the training set (e.g. the object is in a slightly different position or the lighting is slightly changed), ALT will likely retrieve a demonstrator state that is ``close enough” and use the appropriate action from that neighbor. For instance, OOD6-OOD7 in Fig 3 shows different lighting conditions, while OOD1-OOD3 show the presence of distractors in a cluttered environment. In such cases, ALT is typically able to detect OOD inputs and issue appropriate warnings; or do the correct actions. For more general OOD situations which is caused by the mismatch between the wrist camera and the third-view camera, such as in our reactive test (3:39-4:25 in the video), ALT also demonstrates strong generalization performance in handling such OODs. Furthermore, for standard small-scale robot datasets (with approximately 100-200 trajectories), ALT can also handle them, showing a 100\% success rate in InD cases while exhibiting the multimodality characteristic of diffusion policies (4:25-4:49 in the video). If the scenario is outside anything seen (for example, a drastically new background or a totally new object), ALT, like any memory-based method, may not have a good match to draw on. In those cases, an important advantage of ALT is its explicit out-of-distribution (OOD) detection mechanism. We set a similarity threshold in the latent space; if no stored observation is sufficiently close to the current input, ALT will recognize it as OOD. In practice, this means ALT can signal uncertainty or default to a safe fallback when faced with something truly novel. This kind of built-in caution is harder to obtain with a learned Diffusion Policy, which will always produce an action for any input. Regarding the lookup table size: as we add more demonstrations, ALT's coverage of the state space increases. If we had a much larger demonstration set, ALT's raw performance would likely improve. The trade-off is that a larger table means higher memory usage and potentially slower lookup, issues which can be addressed as discussed in our response to your next comment.
> >
> > In summary, ALT performs well as far as its stored data allows: small changes are handled well by finding the nearest known state, and drastic unseen changes are detected via the latent distance. This behavior, we believe, mirrors what Diffusion Policy itself is doing internally (finding the closest training example in latent space), except ALT makes that process explicit and observable.
> >
> > Our response to Questions #2:
> >
> > Our current study operated in the realm of tens of demonstrations per task. Of course, we acknowledge that if the number of demonstrations were scaled up dramatically (say to thousands or more), a naive ALT implementation would face challenges. The latency of naive lookups grows with the number of entries, and storing thousands of high-dimensional embeddings and actions would increase memory usage linearly. However, there are techniques to handle this growth. For instance, one could use an efficient indexing structure (K-D tree) or approximate nearest-neighbor algorithms to keep lookup times fast even with large databases. These methods can handle millions of points with sub-linear search times on modern hardware. On the memory side, if many demonstrations are redundant or similar, one could cluster or compress the representation (e.g. use quantization techniques) to reduce storage without losing much accuracy. Alternatively, for extremely large datasets, it is instead possible to utilize a hash table data structure to provide a worst-time lookup of O(1) (thereby allowing efficient querying) at the cost of having a degraded action recall from hash clashes. We deliberately kept ALT simple (a direct index of all examples) to emphasize transparency, but it’s not fundamentally incompatible with these optimizations.
> >
> > In a multi-task scenario, additional considerations arise: the policy would need to know which task or context it’s in so that it doesn’t retrieve an action from the wrong task. This could be handled by partitioning the memory by task or adding the task identifier as part of the lookup key. Although we did not implement multi-task ALT in this work, conceptually it could be extended (with appropriate context conditioning) if one wanted a non-parametric approach across tasks.

---

> > > ### Author Response · Authors · 2025-11-21
> > > **Reponse to Reviewer aGTM - Part 4**
> > >
> > > Our response to Questions #3:
> > >
> > > You raise an important point. Tasks that require long-horizon reasoning, contact-rich dynamics, or multi-stage planning often involve observation–action dependencies that span beyond a single visual frame. In such settings, a pure lookup-based policy like ALT may face challenges—however, this limitation is not unique to ALT. Our results show that in the low-data regime, Diffusion Policy behaves nearly equivalently to ALT. Therefore, if ALT struggles in these more complex scenarios, Diffusion Policy and other visuomotor policies would likely struggle as well unless provided with substantially more training data.
> > >
> > > It is also plausible that with significantly larger datasets, ALT could retain performance comparable to DP while remaining simpler to implement, debug, and interpret. Investigating such scalability questions—how both ALT and DP behave in richer, long-horizon domains—would require a dedicated research effort and is outside the scope of our current work. Our paper does not claim that ALT alone solves these challenging settings; rather, our contribution is to show that in the low-data, short-horizon tasks studied, a memory-based retrieval strategy can match the performance of diffusion policies while being considerably more lightweight. Exploring hybrid or hierarchical extensions of ALT to handle more complex tasks is an exciting avenue for future research.
> > >
> > > Our response to Questions #4:
> > >
> > > We agree with the reviewer that ALT's performance is dependent on the quality of the learned representation. If we were to use an inferior representation (for example, raw pixel values or an untrained encoder), ALT would indeed perform much worse. We actually validated this by comparing to the naive kNN baseline: using a simple KD-tree on raw image features (or other non task-aligned features) led to less satisfactory results. On the other hand, if we improve the representation, ALT benefits directly.
> > >
> > > In summary, ALT is as good as the features it operates on: we found it to be insensitive to minor encoder design choices (e.g. we also tried a variant including end-effector pose data and saw similar performance), but sensitive to overall representation quality. A better encoder yields better retrievals. We will add discussion of the representation ablation to highlight how much it matters.
> > >
> > > To further understand the role of representation quality in ALT, we evaluated the system under five different encoder architectures—ResNet-18, SimpleCNN, ViT, CLIP, and Swin—while varying their feature dimensions. The results are shown in Table 2. First, ResNet-18 achieves uniformly high performance across all dimensions, reflecting the stability and strong inductive biases of convolutional networks, which are well suited for manipulation-oriented visual inputs. In contrast, SimpleCNN and ViT perform significantly worse: the former lacks sufficient representational capacity, while the latter requires large-scale pretraining. CLIP produces strong features but is highly sensitive to output dimensionality—dimensions that are too small under-represent the feature manifold, whereas excessively large dimensions distort it, leading to degraded contrastive alignment. Finally, Swin exhibits a clear performance improvement as dimensionality increases. Overall, these results highlight that encoder choice and representation quality have a substantial impact on ALT performance, confirming that representation quality is a critical factor for contrastive training.
> > >
> > > **Table 2**
> > >
> > > | Encoder-Dim   | InDs    | InD-1 | OOD1 | OOD2 | OOD3 | OOD4 | OOD5 | OOD6 | OOD7 |
> > > |---------------|---------|-------|------|------|------|------|------|------|------|
> > > | ResNet-64     | 100%    | ✓     | ✓    | ✓    | ✓    | ✗    | ✗    | ✓    | ✓    |
> > > | ResNet-128    | 100%    | ✓     | ✓    | ✓    | ✓    | ✗    | ✗    | ✓    | ✓    |
> > > | ResNet-256    | 100%    | ✓     | ✓    | ✓    | ✓    | ✗    | ✗    | ✓    | ✓    |
> > > | SimpleCNN-64  | 19.35%  | ✗     | ✗    | ✗    | ✗    | ✗    | ✗    | ✗    | ✗    |
> > > | SimpleCNN-128 | 22.58%  | ✗     | ✗    | ✗    | ✗    | ✗    | ✗    | ✗    | ✗    |
> > > | SimpleCNN-256 | 22.58%  | ✗     | ✗    | ✗    | ✗    | ✗    | ✗    | ✗    | ✗    |
> > > | ViT-64        | 12.9%   | ✓     | ✓    | ✓    | ✓    | ✓    | ✓    | ✓    | ✗    |
> > > | ViT-128       | 12.9%   | ✓     | ✓    | ✓    | ✓    | ✓    | ✓    | ✓    | ✓    |
> > > | ViT-256       | 16.13%  | ✗     | ✗    | ✗    | ✗    | ✗    | ✗    | ✗    | ✗    |
> > > | CLIP-64       | 22.58%  | ✓     | ✓    | ✓    | ✓    | ✗    | ✗    | ✗    | ✗    |
> > > | CLIP-128      | 100%    | ✓     | ✓    | ✓    | ✓    | ✗    | ✗    | ✗    | ✗    |
> > > | CLIP-256      | 51.61%  | ✓     | ✓    | ✗    | ✓    | ✓    | ✓    | ✗    | ✗    |
> > > | Swin-64       | 77.42%  | ✓     | ✓    | ✓    | ✓    | ✗    | ✗    | ✓    | ✓    |
> > > | Swin-128      | 83.87%  | ✓     | ✓    | ✓    | ✓    | ✓    | ✗    | ✓    | ✓    |
> > > | Swin-256      | 90.32%  | ✓     | ✓    | ✓    | ✓    | ✗    | ✗    | ✓    | ✓    |

---

### Meta-Review · Area_Chair_zKZj · 2026-01-07

**Summary:**

The initial reviews are very much divergent (8-8-4-2) and unfortunately there's no follow-up discussions from the reviewers. On one hand, the concerns raised by reviewers aGTM and zjbw are centered around the fact that the proposed method "ATL" is not novel nor scalable, and the experiments are too simple. On the other hand, the positive reviewers praised the soundness and clarity of the analysis provided in the paper, as well as serious and structured experimental design, both points combined offered valuable insights on memorization.

Upon reading the paper, the positive reviews resonate with me a lot more. The point of the paper IS NOT to propose a SOTA method for scalable robot learning, nor is this a requirement for a good paper. The authors never claimed that the proposed ATL method should be the future of robotics but rather a hypothesis to understand the underlying mechanism of methods like diffusion policies. I belive the authors have achieved their goals, though with limitations as suggested by the reviewers. Overall, I believe the paper should be accepted.

**Reviewer Concerns:**

Addressed concerns:
- sensitivity to visual representation: the authors added Table 2, an ablation study comparing five different encoders (ResNet, SimpleCNN, ViT, CLIP, and Swin) across various dimensions.
- scalability to large datasets: the authors have acknowledged that the method is not scalable but can provide insights and inspiration to design future algortihms which I agree.

Outstanding concerns:
- limited experimental scope: authors have added extensive new evaluations but are all in simulation. The overall experiments still remain relatively weak.
- applicability to complex and long-horizon tasks remain limited.

**Reviewer Scores:**

In this case, it's really hard to predict whether the reviewers who gave 2 and 4 will change their opinions but my personal opinion is leaning towards accept. And given the existing two strongly positive reviews, I believe the paper should be accepted.

---

### Decision · Program_Chairs · 2026-01-26

Accept (Poster)